# Neural Episodic Control with State Abstraction

**Zhuo Li**[1]    **Derui Zhu**[2]    **Yujing Hu**[3]    **Xiaofei Xie**[4]    **Lei Ma**[5,6],
**Yan Zheng**[7]    **Yan Song**[3]    **Yingfeng Chen**[3]    **Jianjun Zhao**[1]
[1]Kyushu University    [2]Technical University of Munich    [3]NetEase Fuxi AI Lab
[4]Singapore Management University    [5]University of Alberta
[6]The University of Tokyo    [7]Tianjin University
`ma.lei@acm.org    zhao@ait.kyushu-u.ac.jp`

## Abstract

Existing Deep Reinforcement Learning (DRL) algorithms suffer from sample inefficiency. Generally, episodic control-based approaches are solutions that leverage highly-rewarded past experiences to improve sample efficiency of DRL algorithms. However, previous episodic control-based approaches fail to utilize the latent information from the historical behaviors (*e.g.*, state transitions, topological similarities, *etc.*) and lack scalability during DRL training. This work introduces Neural Episodic Control with State Abstraction (NECSA), a simple but effective state abstraction-based episodic control containing a more comprehensive episodic memory, a novel state evaluation, and a multi-step state analysis. We evaluate our approach to the MuJoCo and Atari tasks in OpenAI gym domains. The experimental results indicate that NECSA achieves higher sample efficiency than the state-of-the-art episodic control-based approaches. Our data and code are available at the project website[1].

## 1    Introduction

Deep reinforcement learning (DRL) has garnered much attention in both research and industry, with applications in various fields related to artificial intelligence (AI) such as games (Mnih et al., 2013; Silver et al., 2018; Shen et al., 2020), autonomous driving (Xu et al., 2020), software testing (Zheng et al., 2019; 2021c) and robotics (Thomaz & Breazeal, 2008). DRL usually achieves excellent performance for many tasks and sometimes outperforms human beings. However, human-level DRL policies usually require a tremendous amount of data and millions of training steps, which are demonstrated to be sample inefficient (Arulkumaran et al., 2017; Tsividis et al., 2017). To mitigate this problem, many approaches have been proposed, such as improving the exploration (Yu, 2018; Burda et al., 2018), modeling the environment (Moerland et al., 2020), state abstraction (Vezhnevets et al., 2017) and knowledge transfer (Lazaric et al., 2008; Zhang et al., 2020; Cao et al., 2022). However, this paper focuses on resolving the problem of sample inefficiency through episodic control.

Episodic control is designed to assist DRL agents in making the appropriate decisions in unseen environments using past experiences. The idea is inspired by a biological mechanism, hippocampus (Lengyel & Dayan, 2007). Moreover, episodic control has been adopted to tackle the sample inefficiency in DRL (Blundell et al., 2016; Pritzel et al., 2017). Previous neural episodic control-based approaches usually store past experiences in a tabular memory. Therefore, the agent could retrieve historical highly-rewarded experiences by looking up similar cached states from the episodic memory. Then the state (action) values could be estimated based on the similar states retrieved. In this way, the policy can efficiently reduce the bias between episodic and model estimated state values and generalize the past highly-rewarded cases.

Although many episodic control-based approaches were proposed to improve the sample efficiency of DRL policy, all of them suffer from obvious limitations (Hu et al., 2021; Pinto, 2020; Kuznetsov & Filchenkov, 2021). In general, they only store the concrete states, actions, and state values (Blundell

---

[1]`https://sites.google.com/view/drl-necsa`

et al., 2016). On the other hand, their episodic memory does not record information such as time steps and transitions in the traces. As a result, some latent semantics, such as state transitions and topological similarities, cannot be explored and exploited. However, many previous works demonstrate that such latent information can be used to improve sample efficiency (Kuznetsov & Filchenkov, 2021; Zhu et al., 2020). For instance, a DRL model is trained to make decisions continuously, and the influence (*e.g.*, approximation errors) of a state-action pair might be accumulated and affect the forward scenes (Dynkin, 1965). In other words, the root cause of a *bad* decision in the current state can either come from the latest state or the much earlier states. The topological state transitions can help trace the root cause of the *bad* decisions.

Moreover, the data structure of existing episodic memories does not efficiently support storing and exploring latent semantics. The reason is that the concrete state representations usually consist of float numbers. Thus almost none of the states are the same. Consequently, we cannot directly count and retrieve them from episodic memory, and it is impossible to identify the critical state transitions. In addition, existing episodic control-based approaches, which utilize distance-based measurement to retrieve the $k$-st most similar concrete states (*e.g.*, k-nearest neighbors, *i.e.*, *kNN* search) and the weighted sum of the state (action) values for the estimation (Pritzel et al., 2017; Lin et al., 2018), are inevitably resource-consuming. Overall, existing episodic control-based approaches lack (1) a more comprehensive episodic memory analysis (*i.e.*, multi-step state transitions) and (2) a more scalable storage and retrieval strategy for episodic data.

We propose NECSA, a state abstraction-based neural episodic control approach that enables a more comprehensive analysis of episodic data and better sample efficiency to address the above issues. Inspired by multi-grid and model-based reinforcement learning (Grześ & Kudenko, 2008; Kaiser et al., 2019), we discretize the continuous state space into finite grids on each dimension, and the states located in the same grid will be labeled with a unique *ID*. Naturally, we conduct a multi-step analysis of the state transitions by treating the consecutive state transitions as a fixed pattern. Finally, we make the policy generalize different patterns, as we infer that analyzing and generalizing such multi-step patterns might result in better performance than just focusing on a single state (Sutton & Barto, 2018). Such abstraction enables the following strengths: (1) based on the abstracted state space, more advanced semantic characteristics, such as state transitions and topological similarities, can be analyzed for improving performance (Grześ & Kudenko, 2008); (2) the complexity of storing and retrieving the episodic data is reduced $\mathcal{O}(N)$ to $\mathcal{O}(1)$ since we can retrieve the episodic memory using an exact match.

Previous episodic controls used average state values as the state measurement to correct the DRL policy estimation (Lin et al., 2018; Kuznetsov & Filchenkov, 2021). Nevertheless, such a state measurement cannot be computed directly for a multi-step pattern. Instead, we propose an intrinsic state measurement based on state abstraction instead of past state values. Specifically, we record the returns of those episodes where they occur in an abstract pattern. Then we compute the average returns to measure the abstract pattern. This measurement can efficiently identify those patterns which can result in higher rewards. By utilizing such intrinsic rewards (Burda et al., 2018), we revise the policy and accelerate the learning by encouraging those states with higher measurements but punishing those with relatively low measurements.

Finally, we evaluate NECSA on MuJoCo (Todorov et al., 2012) and Atari tasks in OpenAI gym (Brockman et al., 2016) domains. The evaluation shows that our approach can significantly improve the sample efficiency and outperform state-of-the-art episodic control. In summary, we make the following contributions: (1) we propose a multi-step analysis of state transitions to achieve better policies; (2) we propose a comprehensive episodic memory that enables a more advanced analysis of past experiences; (3) we propose an intrinsic reward-based episodic control method to optimize the policy.

## 2 RELATED WORK

### 2.1 NEURAL EPISODIC CONTROL

Episodic control (Lengyel & Dayan, 2007) was creatively applied on model-free DRL tasks to retrieve episodic memory-based state values (Blundell et al., 2016) for resolving sample inefficiency. The distance-based measurements were applied for looking up similar episodic data (Pritzel et al., 2017). The episodic memory buffer can be smaller by applying Gaussian Random Projection to reduce the

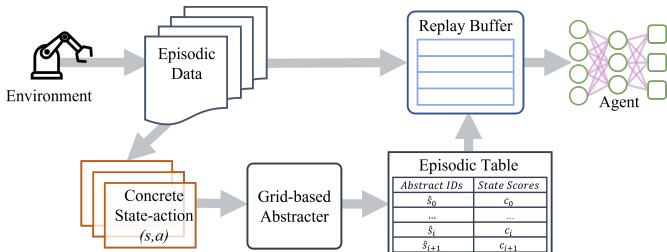

Figure 1: The overview of NECSA. We abstract the concrete state-action pairs, measure the abstract states and combine the state measurement with the traces. Finally, the revised traces will be stored in the replay buffer for the agent to sample.

dimension of concrete states (Lin et al., 2018). The retrieved state values from episodic memory can be combined with the predictions of the critic network (Hansen et al., 2018). Novel episodic memory structure such as associative memory and generalized memory efficiently propagates state values to stored memory items (Zhu et al., 2020; Hu et al., 2021). Episodic control was also combined with multi-agent tasks (Zheng et al., 2021a) with curiosity-based exploration and model-based reinforcement learning (Le et al., 2021). Recently, the episodic control was also applied to the continuous control (Zhang et al., 2019; Kuznetsov & Filchenkov, 2021), which outperformed the state-of-the-art DRL algorithms. The universal episodic control (Hu et al., 2021; Pinto, 2020) was proved effective on both discrete and continuous action space by adopting a generalized episodic memory (*e.g.*, neural network) to fit the past state values. Offline tasks can be solved by the state value-based episodic buffer and avoid the over-generalization of actions in the dataset (Ma et al., 2021). Episodic memory was also used to search the optimal hyperparameter for policy gradient methods (Le et al., 2022). This paper proposes NECSA, a novel and simple episodic control-based approach. NECSA adopts a grid-based state abstraction instead of distance-based measurement to operate episodic memory more efficiently. Moreover, unlike previous works, our episodic memory is more comprehensive since the state abstraction strategy enables us to explore abstract state transitions and topological information. Such information can be used to improve the sample efficiency (Yin & Li, 2020).

## 2.2 STATE ABSTRACTION

The purpose of state abstraction is to group states which share the same characteristics into a single cluster. One solution is discretizing the continuous domains (Dougherty et al., 1995). In model-based RL tasks (Kaiser et al., 2019), the state abstraction (Jiang et al., 2015; Burda et al., 2018) plays an essential role in reducing the data scales, although such abstraction requires strong domain knowledge of the environment or assumptions. In model-free reinforcement learning, the grid-based state abstraction has proved effective (Anderson & Crawford-Hines, 1994; Grześ & Kudenko, 2008). Particularly in continuous control, the continuous action space can be divided into equal intervals, then those concrete actions in the same value scales can be labeled by a common abstraction (Pazis & Lagoudakis, 2009; Tang & Agrawal, 2020; Du et al., 2019; Xie et al., 2019; Zhu et al., 2021). Another work of state abstraction in reinforcement learning is DreamerV2 (Hafner et al., 2020) which encodes the observations directly as discrete state variables. The idea of smoothing the state space (Gangwani et al., 2020) shares similarities with our work exploring topological information on state transitions. Dynamic state clustering (Mannor et al., 2004) is proved to be effective, although it is hard to be applied to the multi-step analysis. Moreover, previous works partially focus on state or action space, and few works have been done on abstracting state-action pairs. Inspired by the above works, we use the grid-based abstraction to group the state-action pairs simultaneously.

## 3 BACKGROUND

Generally, the process of reinforcement learning (RL) can be defined as a Markov Decision Process (MDP) (Sutton & Barto, 2018). MDP is a four-tuple $\langle S, A, R, P \rangle$, where $S$ and $A$ represent the sets of states and actions respectively. The agent interacts with the environment at each time step $t$ by observing the current state $s_t \in S$, choosing an action $a_t \in A$ to execute, receiving an immediate

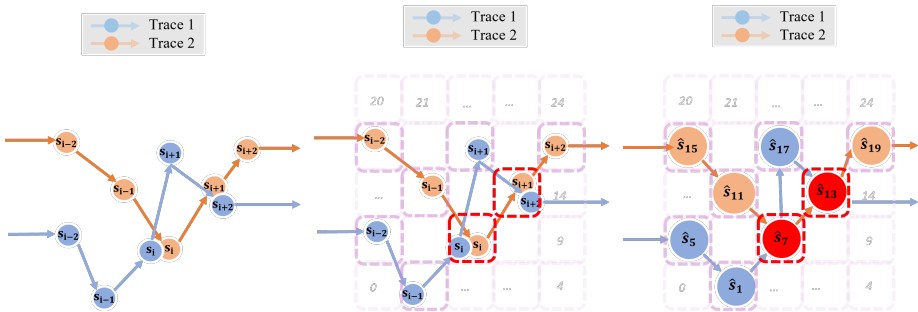

(a) Concrete state transitions. (b) Grid-based state abstraction. (c) Abstract state transitions.

Figure 2: Take a 2-dimensional concrete state space as an example. Concrete states can be labeled by the *ID* of grids. For example, In trace 1, state $s_{i-1}$ is abstract state $\hat{s}_1$. Concrete state $s_{i+2}$ in trace 1 and $s_{i+1}$ in trace 2 are located in the same grid; thus, they share the same abstract state $\hat{s}_{13}$ (red ones). In this way, trace 1 can be represented as $(\hat{s}_5, \hat{s}_1, \hat{s}_7, \hat{s}_{17}, \hat{s}_{13})$.

reward $r_t = R(s_t, a_t)$ after doing $a_t$, and transferring to a new state $s_{t+1} \sim P(s_t, a_t)$. The $R(\cdot)$ and $P(\cdot)$ are the reward function and state transition function respectively.

The agent selects an action $a \sim \pi(s)$ to execute according to a policy $\pi(\cdot)$ and interacts with the environment to generate a trace, $[(s_0, a_0, r_0), ..., (s_t, a_t, r_t), ...]$, where the subscripts denote different time steps. The return of each trace is defined by $\sum_{t=0}^{T} \gamma^t r_t$, where rewards are discounted by a factor $\gamma \in [0, 1)$. RL generally aims to find an optimal policy to maximize the returns.

The state-action value function: $Q_\pi(s_t, a_t) = E_\pi\big[\sum_{t=0}^{T-1} \gamma^t r_t | s_t, a_t\big]$ is widely used in many RL algorithms. In value-based approaches (e.g., DQN (Mnih et al., 2013)), action with maximal $Q_\pi(s_t, a_t)$ value will be selected, and these approaches have achieved great success in discrete action space environments. The policy-based approaches such as A3C (Mnih et al., 2016) are more efficient and suitable for continuous action spaces than value-based approaches, which generate a policy distribution and sample actions from it instead of iterating in the infinite action space. DDPG (Lillicrap et al., 2016) is used in a continuous action setting to improve the sample efficiency over the vanilla actor-critic, in which the *deterministic* means the Actor can directly output the actions instead of computing a probability distribution over actions. Twin Delayed DDPG (TD3) (Fujimoto et al., 2018) tackles the over-estimation of state values. In this work, we build our episodic control in continuous action space based on TD3 (Fujimoto et al., 2018).

## 4 METHODOLOGY

This paper proposes an effective and efficient episodic control framework, NECSA. Figure 1 shows the main procedure of our framework. We have two additional modules: (1) a grid-based abstraction to convert the concrete states to abstract *ID* (*i.e.*, the $\hat{s}_i$ as noted later); (2) a key-value-manner episodic memory module to store the abstract states with the state measurement (*i.e.*, the reward confidence scores $c_i$ as mentioned later). The traces are revised based on the state measurement and further flow to the replay buffer for policy optimization. NECSA is highly supplementary, which can be applied to the general reinforcement learning paradigm.

### 4.1 GRID-BASED STATE ABSTRACTION

The $K$-dimensional state space $R^K$ contains infinite concrete states, each state $s_i$ is represented as $(s_i^0, ..., s_i^K)$. Each component $s_i^m$ in the $m$-th dimension satisfies $s_i^m \in [l_m, u_m]$, where the $l_m$ and $u_m$ are the lower and upper boundaries, respectively. We split each dimension $[l_m, u_m]$ into $N$ equal intervals. Thus the concrete state space $R^K$ will be divided into $N^K$ grids as follows:

$$e_n^m = [l_m + n \times \frac{u_m - l_m}{N}, l_m + (n+1) \times \frac{u_m - l_m}{N}] \tag{1}$$

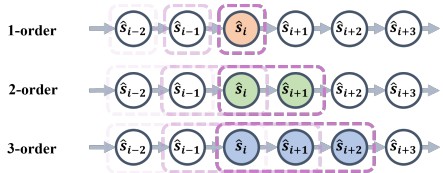

Figure 3: A schematic diagram of the multi-step analysis. We slide windows with different lengths to extract patterns.

where $e_n^m$ represents the $n$-th interval on the $m$-th dimension in the state space. In this way, (1) each concrete will fall into a grid; (2) all the concrete states $s_i$ which fall into the same grid can be identified as in the same cluster; (3) each cluster (*i.e.* each grid) can be represented by a unique abstract state $\hat{s}$:

$$\hat{s} = \{s_i | s_i^m \in e_n^m, n \in [0, N], m \in [0, K]\} \tag{2}$$

Figure 2 is a schematic diagram of grid-based clustering, in which the infinite concrete state space is converted into the finite discrete state space. The environment determines the $l_m$ and $u_m$. It is worth mentioning that we can either perform the abstraction on concrete states or state-action pairs. As the experimental results suggest, abstracting state-action pairs can perform better than focusing on concrete states. Take Walker2d-v3 as an example; the state vector's shape is (1,17), and the action vector is (1,6). Then, we combine the state and action vectors into a state-action pair with a shape of (1,23). The boundaries of the state and action vectors are [-10,10] and [-1,1], respectively. For instance, we obtain the fixed lower bounds as (-10, -10, ..., -10, -1, -1, ..., -1). finally, we abstract the state-action pairs by grid-based clustering. Each dimension of a state-action pair is split into equal intervals.

In general, the high-dimensional concrete state (-action pairs) would increase the difficulty and complexity of state abstraction. Inspired by Gaussian random projection (Dasgupta, 2013), we apply a random matrix to reduce the dimensions of state (-action) vectors. When creating the matrix, we ensure that the values of the matrix elements follow the Gaussian distribution and that the value scope is [-1,1]. By multiplying the state (-action) vector (*e.g.*, (1,376) in Humanoid-v3) to a Gaussian random matrix with a shape of (376,24), we can obtain a smaller state (-action) representation vector as (1,24). The Gaussian random matrix is initialized at the beginning of the training. Then, it would remain the same during the training. Finally, all the state (-action) vectors are projected to a smaller one by a common Gaussian random matrix.

Note that abstracting images directly can be inefficient in Atari games. Pixel-based image vectors consist of multiple channels, which can be unsuitable for abstraction using grid-based methods. Therefore, we use the hidden outputs for abstraction, considering that (1) the hidden outputs are natural features of images but relatively smaller than the raw images, and (2) hidden outputs reflect the states of actions. Moreover, incorporating hidden outputs with episodic control is also effective in previous episodic control approaches such as EVA (Hansen et al., 2018).

## 4.2 EPISODIC MEMORY WITH MULTI-STEP ANALYSIS AND STATE MEASUREMENT

Figure 3 shows the logic of extending the state abstraction to multi-step patterns. The abstract patterns are extracted by sliding a $N$-step window on each trace. We treat $N$-step continuous abstract states as a whole pattern. Moreover, we count the patterns iteratively and orderly, where $\{\hat{s}_i, \hat{s}_{i+1}, \hat{s}_{i+2}\}$ and $\{\hat{s}_{i+1}, \hat{s}_{i+2}, \hat{s}_{i+3}\}$ are treated as different patterns.

To revise the state (action) values (*i.e. Q*) estimation, previous episodic control-based approaches usually leverage a shift of episodic returns to correct the critic Q-value. Such approaches can achieve better performance while requiring domain-specific knowledge of corresponding DRL algorithms, model structure reform, and engineering work efforts. Furthermore, measuring states by the statistics of Q-values is only applicable in one-step episodic memory. With multi-step patterns, we propose a state measurement named reward confidence scores. The score $c_i$ represents the expectation of the episode return when $\hat{s}_i$ occurs. Although the scores are similar to state (-action) values in terms

of the definitions, they stand for different semantics. State (action) values represent the forwarding rewards after the state $\hat{s}_i$, but the scores measure the episode reward over the whole trace.

We design a simple key-value-manner episodic memory $C$ based on state abstraction and measurement. Each item in the episodic memory is indexed by the $\hat{s}_i$ with three values: the current score $c_i$, the historical total episodic rewards $\mathcal{E}_i$ earned by the traces which contain the $\hat{s}_i$, and the total number of occurrences $\eta_i$ of $\hat{s}_i$. The following equation is the implementation of computing scores:

$$\begin{cases} \mathcal{E}_i = \sum r, & \eta_i = 1, & c_i = \mathcal{E}_i, & \text{if } \hat{s}_i \notin C \\ \mathcal{E}_i = \mathcal{E}_i + \sum r, & \eta_i = \eta_i + 1, & c_i = \dfrac{\mathcal{E}_i}{\eta_i}, & \text{otherwise,} \end{cases} \tag{3}$$

Where the $\sum r$ is the final episodic return of the trace, which contains $\hat{s}_i$. In this way, our memory buffer can efficiently record past experiences and update the scores. Our episodic memory enables three operations: *ADD*, *LOOKUP*, and *UPDATE*. *ADD* is the operation to store the $\hat{s}_i$ into $C$. *LOOKUP* returns score $c_i$ for $\hat{s}_i$ by taking $\mathcal{O}(1)$. *UPDATE* is to revise the $c_i$ of $\hat{s}_i$. Moreover, the episodic memory enables the normalizing of the scores to [0,1], and the average of the score $c_i$ is adopted as the intrinsic reward for abstract pattern $\hat{s}_i$.

---

**Algorithm 1** NECSA.

---

**Input:** $\mathcal{M}$: DRL model, $E$: Environment, $\mathcal{B}$: Replay Buffer
**Input:** $C$: An episodic memory to record each $\hat{s}_i$ and the score $c_i$
**Output:** $\mathcal{M}'$: A tuned DRL model
1: Initialization: $i \leftarrow 0, S, \hat{S}, A, R, D, \mathcal{B} \leftarrow \emptyset$
2: **while** $i < Total\_steps$ **do**
3:      $s_i \leftarrow state, a_i \leftarrow action, r_i \leftarrow reward, d_i \leftarrow done$
4:      $S.append(s_i), A.append(a_i), D.append(d_i)$
5:      $\hat{s}_i \leftarrow State\_Abstract(s_i, a_i)$
6:      $\hat{S}.append(\hat{s}_i)$
7:      $c_i \leftarrow Inquire\_Score(\hat{s}_i, C)$
8:      $\hat{r}_i \leftarrow Reward\_Revision(r_i, c_i)$
9:      $R.append(r_i)$
10:     $\mathcal{B}.add(s_i, a_i, \hat{r}_i)$
11:     **if** $d_i$ is true **then**
12:        $C \leftarrow Update\_Score(\hat{S}, R, C)$
13:        $S, \hat{S}, A, R, D \leftarrow \emptyset$
14:     **end if**
15:     $i \leftarrow i + 1$
16:     $\mathcal{M} \leftarrow Train(\mathcal{M}, \mathcal{B}.sample())$
17: **end while**

---

### 4.3 EPISODIC CONTROL-BASED LEARNING

As inspired by previous episodic control-based approaches, we infer that a statistic-based measurement of state-action pairs can be a helpful reference to revise the estimation of state (-action) values by the critic network (Lin et al., 2018; Kuznetsov & Filchenkov, 2021). The core of our method is to measure the abstract patterns by scores and make the agent generalize the high-score experiences. Previous work like reward shaping (Harutyunyan et al., 2015; Laud & DeJong, 2003; Devlin & Kudenko, 2012; Burda et al., 2018) helps the policy incorporate domain knowledge into policy optimization. Therefore, it is natural to reshape the reward with a score-based intrinsic reward to revise the policy. Formally, the revised reward $\hat{r}_i = r_t + \Delta$, where $r_t$ is the return of the original reward function, $\Delta$ is the intrinsic reward (computed based on the reward confidence scores) which represents the semantics for improving the rewarding mechanism. We perform reward revision for each pattern as follows:

$$\hat{r}_i = r_i + \underbrace{(c_i - \frac{\sum_{m=0}^{M} c_m}{M})}_{\Delta} \times \epsilon, \tag{4}$$

where the $\hat{r}_i$ is the revised reward based on $r_i$. $M$ is the total number of abstract patterns. $\frac{\sum_{m=0}^{M} c_m}{M}$ is the average of all the scores. The term $\Delta$ will be less than 0 if the $\hat{s}_i$ score is less than the average. Thus the corresponding concrete state will be punished. The state will be encouraged if $\Delta$ is greater than 0. The agent will earn more rewards if it passes through $\hat{s}_i$. In a word, the lower the score $c_i$ is, the more the pattern is punished. $\epsilon$ is a hyperparameter that helps control the magnitude of punishment and encouragement.

We implement NECSA as Algorithm 1. We use a module named *State_Abstract* in line 5, which can covert the concrete states to abstract patterns in run-time. While tuning the model, scores can be inquired from the episodic memory $C$ in line 7, and the reward will be revised in line 8. We also added a module named *Update_Score* in line 12, which helps calculate the scores for the abstract patterns in $\hat{S}$ and update the scores in $C$ as each termination of an episode. The revised rewards will be stored in the replay buffer $\mathcal{B}$. Finally, the policy will be updated with the sampled data from the $\mathcal{B}$ in line 16.

## 5 EXPERIMENT

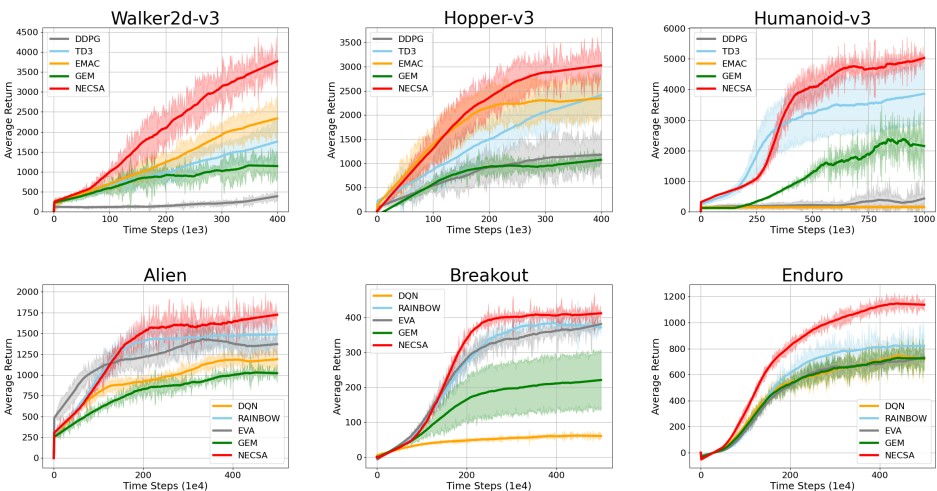

Figure 4: The evaluation of each approach on MuJoCo and Atari tasks. More evaluation results are in Appendix.

### 5.1 EXPERIMENT SETUP

We conduct the experiments on nine MuJoCo tasks and six Atari games in OpenAI gym (Brockman et al., 2016) domains. For continuous control tasks in MuJoCo, we select four baselines including EMAC (Kuznetsov & Filchenkov, 2021), GEM (Hu et al., 2021) DDPG (Lillicrap et al., 2016) and TD3 (Fujimoto et al., 2018) for comparison. We also compare NECSA to four baselines, including DQN (Mnih et al., 2013), Rainbow (Hessel et al., 2018), EVA (Hansen et al., 2018) and GEM (Hu et al., 2021) on Atari games, which use image-based states and discrete action spaces. EMAC is a state-of-the-art episodic control approach that proved effective on continuous control tasks. EVA is effective on discrete action spaces by performing ephemeral adjustments of Q-values to impact the parametric value functions. GEM integrates a neural network to generalize episodic experiences, enabling fast episodic memory retrieval. GEM can be applied to both continuous and discrete action spaces.

### 5.2 EVALUATION RESULTS

Figure 4 shows the main evaluation results of each approach on MuJoCo and Atari tasks. Note that more evaluation results are in the Appendix A.2. In our experiments, we set the length of the multi-step abstract pattern $m$=3. As shown in Figure 4, our approach NECSA significantly

outperforms the baselines on all tasks. Table 3 and Table 4 in the Appendix are the average returns and standard deviations of the learned policies over multiple random seeds. Overall, we found that (1) NECSA is the most sample-efficient algorithm on various tasks, while other algorithms only establish good performance on the part of tasks; (2) NECSA performs better on relatively complex tasks (*e.g.*, high-dimensional states and image-based states). Such results prove that NECSA is a scalable episodic control on continuous and discrete control tasks.

Take Humanoid-v3 as an example, a 376-dimensions state space with five grids on each dimension generates a $5^{376}$ abstract state space, which is impossible to efficiently store and retrive in the episodic memory. On Atari games, we take the hidden outputs after the convolutional neural network layers of the policy network to represent the image states (-action pairs). However, the dimension of hidden outputs is also unsuitable for grid-based abstraction. Gaussian random projection can be a promising solution to make the concrete states (or hidden outputs) smaller (Pritzel et al., 2017; Kuznetsov & Filchenkov, 2021). Therefore, we perform a dimension reduction of the concrete state vectors to $1 \times 24$ by a random projection and divide each dimension into fixed intervals. The results show that the random projection reduces dimension without significant side effects, thus making NECSA a scalable episodic control on high-dimensional continuous control and image-states tasks.

The experimental results on MuJoCo tasks and Atari games prove that NECSA is effective on various DRL tasks. We performed an in-depth ablation studies on the critical parts of NECSA in the following section. Besides, we have conducted detailed hyperparameter analyses. Note that the hyperparameter analyses are in Appendix A.3. We also proved that NECSA is effective on continuous control tasks with image-based inputs. The related experiments are conducted on DMControl (Tunyasuvunakool et al., 2020) tasks. Please refer to Appendix A.4 for details.

## 5.3 ABLATION STUDY

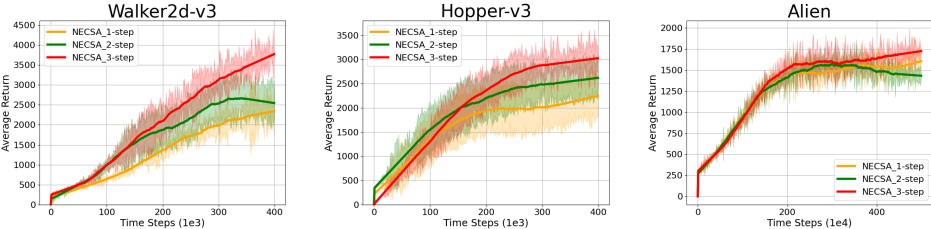

Figure 5: The evaluation of performance between one-step and m-step NECSA.

One of the most critical parts of this paper is to adopt a multi-step analysis of past experiences. To better demonstrate the effectiveness of the multi-step episodic control, we compare the performance between different multi-step settings. The results in Figure 5 show that the three-step NECSA outperforms one-step and two-step settings. In our evaluation, the default setting of NECSA multi-step ($m$) analysis is set to 3. The reason is that longer patterns cause the exact matches to become exceedingly rare, which makes episodic control less helpful for improving performance.

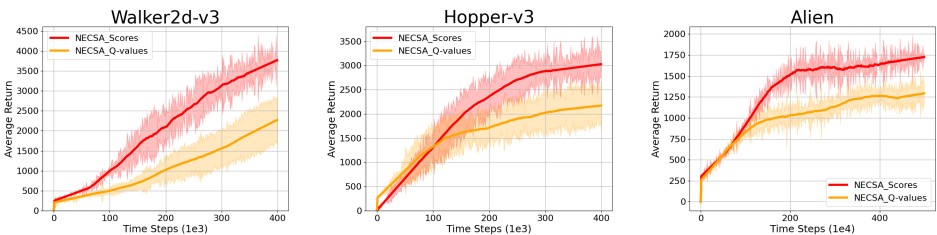

Figure 6: The comparison between measuring the abstract state by scores and Q-values.

We also study the effectiveness of reward confidence scores. We use scores to measure each state-action pair, although previous works mainly focus on the state-action values (*i.e.*, the Q-values). Specifically, we use the average of scores and Q-values (Lin et al., 2018; Kuznetsov & Filchenkov,

2021) for each abstract state (pattern) to compute the intrinsic reward. Note that we cannot directly attain the Q-value of a multi-step abstract pattern. We instead use the average Q value of the corresponding concrete states (Lin et al., 2018). The results in Figure 6 show that measuring the state-action pairs through scores (NECSA_Scores) can lead to better sample efficiency than using Q-values (NECSA_Q-values). We claim that such state measurement is effective and necessary since the scores contain the overall impact of an abstract on a whole trace. Furthermore, such a difference makes the scores more comprehensive since we reference the historical impact on the abstract states.

We conclude that NECSA performs better than the state-of-the-art episodic control-based approaches based on the experiments. State abstraction-based episodic memory enables more advanced analysis of past experiences. The scores of state abstractions are valuable, and the rationale for state measurement and the multi-step analysis of state transitions can significantly improve the performance.

## 6   DISCUSSION

We adopt grid-based abstraction instead of *kNN*. Although *kNN* can group concrete states into unique IDs, it cannot update episodic memory in real-time. Moreover, *kNN* does not support inquiries about the state measurement of multi-step patterns. We extend the episodic control to multi-step analysis, which is one of the most important contributions of NECSA. Overall, we select a grid-based abstraction (clustering) instead of *kNN* for (1) more efficient state abstraction and (2) multi-step abstract pattern analysis.

Although NECSA is sample efficient, there still exists room for improvement. First of all, the state abstraction strategy might cause variances or errors. For example, we divide each dimension by the same number of intervals, though sometimes the value scale of each dimension is not the same. Therefore, some concrete states might not be labeled by abstract *ID*. Based on prior knowledge of the state space, we infer that a possible solution may be to automatically divide each dimension into appropriate but not fixed intervals.

The motivation behind NECSA is to construct a more comprehensive episodic memory. We consider that past experiences need to be further explored, and more valuable information could be obtained and exploited. Furthermore, exploring latent semantics based on the abstraction-based episodic memory promises to improve the sample efficiency. In addition, NECSA is highly scalable and can be applied to various DRL algorithms and tasks.

## 7   CONCLUSION AND FUTURE WORK

We propose NECSA, a novel state abstraction-based neural episodic control approach to improve DRL's sample efficiency. We transform dense concrete state-action pairs into abstract patterns, which enables efficient storage and retrieval of episodic data. We also perform a multi-step analysis of the pattern transitions to enhance their performance. Furthermore, we propose a novel state measurement metric, reward confidence scores, which can be incorporated with extrinsic rewards to accelerate policy optimization. The evaluation results prove that our approach is more sample efficient than state-of-the-art episodic control approaches.

We believe that more advanced research can be conducted based on the state abstraction, state measurement, and multi-step analysis of state transitions, including (1) abstracting the behaviors of the policy via state abstraction since we can build an automaton base on the abstraction; (2) analyzing the state transitions and topological relationships of multi-agent tasks (Zheng et al., 2018; 2021b); (3) applying our approach to model-based reinforcement learning with dynamic planning. We leave these topics for future work.

### ACKNOWLEDGMENTS

This work was supported in part by JSPS KAKENHI Grant No.JP19H04086 and No.JP20H04168, JST-Mirai Program Grant No.JPMJMI20B8, JST SPRING Grant No.JPMJSP2136, as well as Canada CIFAR AI Chairs Program, the Natural Sciences and Engineering Research Council of Canada (NSERC No.RGPIN-2021-02549, No.RGPAS-2021-00034, No.DGECR-2021-00019), the National Natural Science Foundation of China (Grant No.62106172), the "New Generation of Artificial Intelli-

gence" Major Project of Science & Technology 2030 (Grant No.2022ZD0116402), and the Science and Technology on Information Systems Engineering Laboratory (Grant No.WDZC20235250409, No.WDZC20205250407).

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

# A  APPENDIX

## A.1  COMMON HYPERPARAMETER AND SETTINGS

Our hyperparameter settings are listed in Table 1 and Table 2. In addition, we directly evaluate the program released by the authors of EMAC (Kuznetsov & Filchenkov, 2021), EVA (Hansen et al., 2018), and GEM (Hu et al., 2021). We also adopt the implementation of DDPG, TD3, DQN, and Rainbow in (Weng et al., 2021). For MuJoCo tasks, the neural network consists of two hidden layers. The size of each layer is 256. The activation unit is ReLU. Both the Actor and Critic networks share the same structure. For Atari tasks, we use a three-layer convolution neural network head and a fully connected layer to output the Q-values of each action. The specific parameter values are listed in Table 2. More details are available in the code repository.

| Hyper-paramater | NECSA |
|---|---|
| Critic Learning Rate | 3e-4 |
| Actor Learning Rate | 3e-4 |
| Optimizer | Adam |
| Replay Buffer Size | 100000 |
| Batch Size | 100 |
| Discount Factor | 0.99 |
| Episode Length | 1000 |
| Exploration Policy | N(0, 0.1) |

Table 1: Hyperparameters for MuJoCo tasks.

| Hyper-paramater | NECSA |
|---|---|
| Learning Rate | 1e-4 |
| Optimizer | Adam |
| Replay Buffer Size | 100000 |
| Batch Size | 32 |
| Discount Factor | 0.99 |
| Filter Sizes | [8, 4, 3] |
| Filter Strides | [4, 2, 1] |
| Channels | [32, 64, 64] |

Table 2: Hyperparameters for Atari games.

## A.2  THE RESULTS OF NECSA AND OTHER BASELINES ON MUJOCO AND ATARI TASKS.

Table 3: The average returns of learned policies on MuJoCo tasks.

| Task Name | DDPG | TD3 | EMAC | GEM | NECSA |
|---|---|---|---|---|---|
| Walker2d | 387.73±48.34 | 1756.09±240.76 | 2338.24±215.81 | 1224.94±183.48 | **3768.08±270.21** |
| Hopper | 1180.64±276.70 | 2411.15±323.56 | 2347.98±375.97 | 1150.24±138.52 | **3026.57±292.30** |
| Humanoid | 431.69±139.10 | 3858.07±703.91 | 150.32±65.59 | 2380.91±757.82 | **5029.99±208.15** |
| Ant | 811.53±221.40 | 5223.94±270.36 | 789.99±101.04 | 4107.28±254.35 | **5588.44±136.28** |
| HalfCheetah | 11314.36±424.11 | 10120.34±775.47 | 11052.82±669.18 | 11458.70±136.31 | **12097.91±152.09** |
| Swimmer | 96.73±13.10 | 49.82±5.56 | 86.54±22.13 | 126.22±12.70 | **149.63±6.69** |
| Reacher | -4.05±0.14 | -4.01±0.08 | -5.01±0.12 | -5.99±0.39 | **-3.97±0.05** |
| Pendulum | 1000.0±34.77 | 535.97±56.50 | 1000.0±42.50 | 655.48±101.19 | **1000.0±38.17** |
| DoublePendulum | 9316.83±88.85 | 9333.19±226.34 | 9327.17±158.01 | 5858.88±881.50 | **9359.33±263.09** |

We adopt the implementations of DDPG, TD3, DQN, and Rainbow in Tianshou (Weng et al., 2021). The implementation of NECSA is based on TD3. We use the public replicate package of the respective baselines for experiments. We run each experiment with five different random seeds to counteract the randomness and compare the average performance over respective training steps. All the experiments were run on powerful servers with CPU (Intel(R) Core(TM) i9-10940X CPU @ 3.30GHz), and GPU(NVIDIA Corporation GA102GL [RTX A6000]) with 128GB RAM.

We have evaluated NECSA on nine MuJoCo tasks and six Atari games in total. Table 3 and Table 4 are the performance and standard deviation of all the learned policies of the respective approaches. Figure 7 is the performance trends of all the approaches on different tasks (*i.e.*, except the results we have reported in Figure 4). Based on the above results, we found that the learned policies of NECSA achieve the highest performance. Most of the improvements against other approaches are significant. On the tasks such as InvertedPendulum-v2, InvertedDoublePendulum-v2, and Reacher-v2, NECSA reaches the same performance or slightly outperforms the baselines. The reason is that such tasks are relatively easy than others. Therefore NECSA and other baseline approaches can achieve state-of-the-art performance in fewer training steps. Overall, the evaluation results demonstrate that NECSA is effective and sample-efficient on MuJoCo and Atari tasks.

Table 4: The average returns of learned policies on Atari games.

| Task Name | DQN | Rainbow | EVA | GEM | NECSA |
|---|---|---|---|---|---|
| Alien | 1188.73±58.37 | 1488.84±114.93 | 1432.22±109.17 | 1030.67±48.17 | **1725.56±83.03** |
| Breakout | 61.26±4.01 | 385.07±12.33 | 381.29±15.63 | 220.73±54.09 | **412.30±9.94** |
| Enduro | 748.45±49.08 | 824.62±45.77 | 733.50±24.98 | 729.60±49.87 | **1147.48±29.26** |
| MsPacman | 1997.72±120.59 | 2075.16±106.50 | 1991.82±78.71 | 1990.81±134.34 | **2205.28±81.38** |
| Qbert | 9670.13±464.31 | 10817.71±656.88 | 9617.25±464.31 | 9643.69±492.48 | **112659.28±471.96** |
| SpaceInvaders | 629.85±42.62 | 810.12±49.93 | 799.06±26.32 | 686.25±53.36 | **951.62±28.03** |

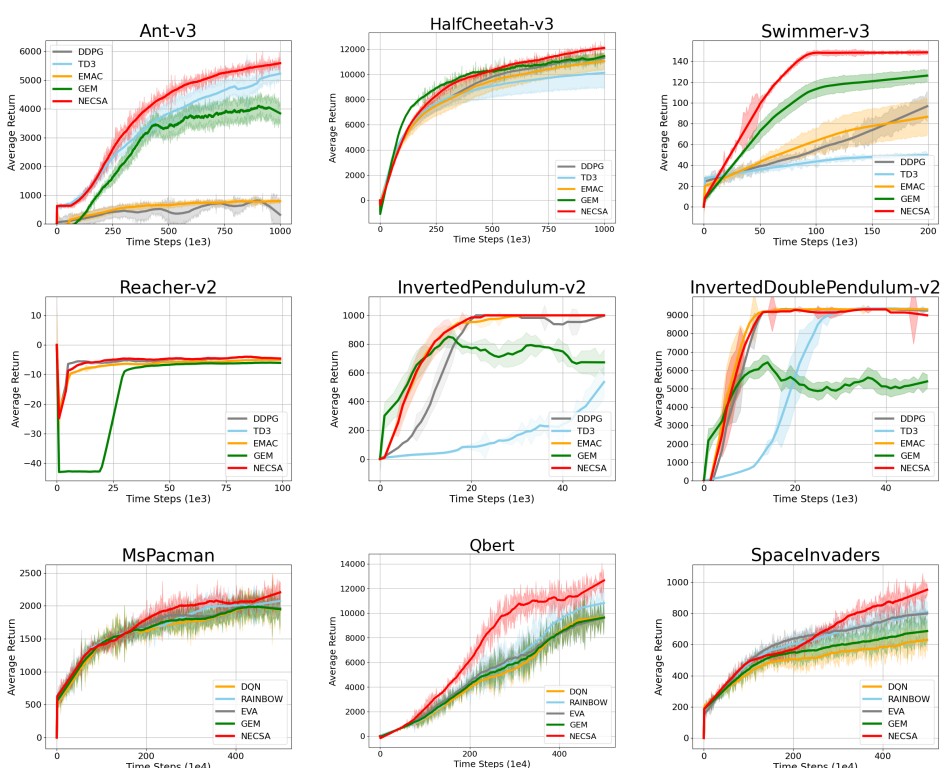

Figure 7: The evaluation of each approach on other MuJoCo and Atari tasks.

## A.3 SPECIAL HYPERPARAMETERS AND SETTINGS IN NECSA

During the experiments, we evaluated different settings of several hyperparameters, which can help us better understand the effectiveness of NECSA. We discuss four settings and answer the following questions through detailed experiments:

1. Should we abstract the state or state-action pairs (or the hidden outputs)?

2. What is the effect of discrete state space grid numbers on performance?

3. How to control the magnitude of revising environment rewards via intrinsic reward (*i.e.*, reward confidence scores)?

4. How do the backbone algorithms affect the performance of NECSA?

Figure 8 shows the performance of abstracting state and state-action pairs. NECSA_state is the curve that we only abstract the concrete states (or the static features of the image states). On the other hand, NECSA_state-action means that we abstract state-action pairs. It reveals that abstracting the state-action pairs can significantly improve performance on MuJoCo tasks. The reason is that abstracting state-action pairs can involve more information for modeling the policy than just focusing on states.

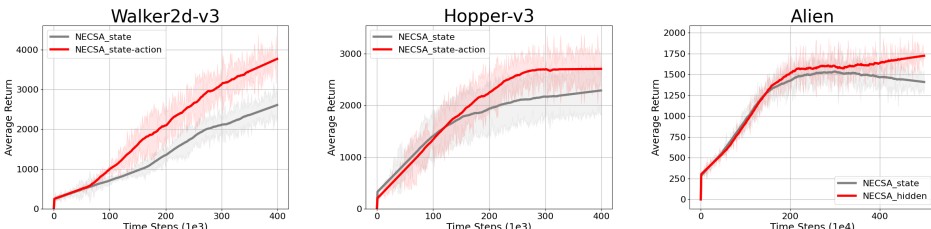

Figure 8: Comparison between abstraction of the state, state-action pairs, and hidden outputs.

NECSA_hidden means that we use the hidden outputs of the policy network for abstracting. Using the hidden outputs on Atari games can also outperform the performance of just abstracting the states. The hidden outputs are natural features of images but relatively smaller than the raw images. Moreover, the hidden outputs reflect the states of actions. Therefore, we infer that using hidden outputs for abstraction shares the same benefits as using state-action pairs.

Figure 9 shows the performance under different abstraction granularities. Grid-N means we divide each dimension of state-action pairs into N equal intervals. The results show that grid numbers 8 and 10 are the best abstraction choices in Walker2d-v3, Hopper-v3 and Alien. Our evaluation is limited to 10 grids since a greater grid number causes the exponential growth of the abstract number, which is very memory-consuming. We found that more grids cannot always achieve better performance. For example, in Walker2d-v3, 8-grids abstraction outperforms 10-grids. The reason is that although more grids can cluster similar concrete states more accurately, as the grid number of the abstraction becomes smaller, the scores of abstract states and patterns are less representative. For MuJoCo tasks, we set the grid numbers to 10 for Swimmer-v3 and Hopper-v3, and 5 for other tasks, respectively. For Atari tasks, we uniformly set the grid numbers to 5.

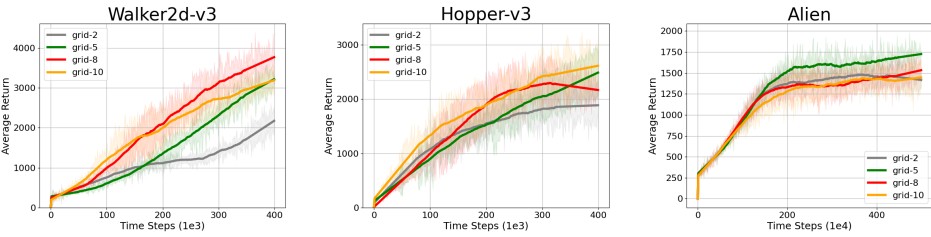

Figure 9: Comparison between different granularities of grid-based abstraction.

Figure 10 shows how the parameter $\epsilon$ in Equation 4 influences the performance. The results suggest two major conclusions: (1) $\epsilon$ in [0.1, 0.2] is the best choice for controlling reward shaping in Walker2d-v3 and Hopper-v3, respectively; (2) a value for $\epsilon$ that is greater or smaller will reduce performance. The reason is that a smaller $\epsilon$ reduces the impact of scores, and a greater $\epsilon$ will increase the weight of state evaluation and completely reform the rewarding mechanism. In addition, for MuJoCo tasks, we set $\epsilon = 0.15$ for Swimmer-v3 and $\epsilon = 0.2$ for other tasks. We set $\epsilon = 0.1$ for all the Atari games.

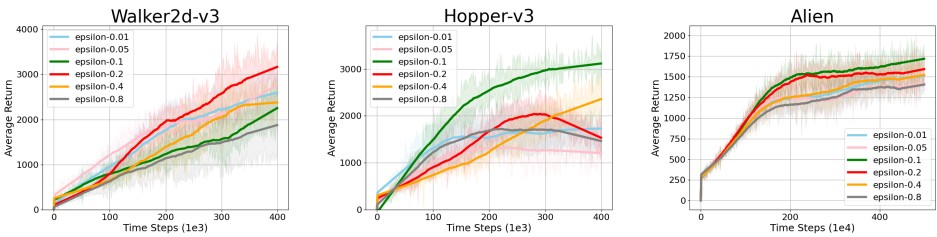

Figure 10: Comparison of performance between different granularities of revising rewards.

Figure 11 shows how the backbone algorithms affect the performance of NECSA. In this evaluation, we build NECSA on TD3 and DDPG and compare the performance with the primary algorithms. The results show that (1) algorithms can achieve better sample efficiency by incorporating with NECSA and (2) the backbone algorithms affect the performance of NECSA. For example, in Walker2d-v3 and Hopper-v3, TD3 is more sample efficient than DDPG. In Alien, Rainbow outperforms DQN. After applying NECSA to these algorithms, the comparison results remain the same trend. Such a conclusion demonstrates that NECSA is scalable and effective on different backbone algorithms.

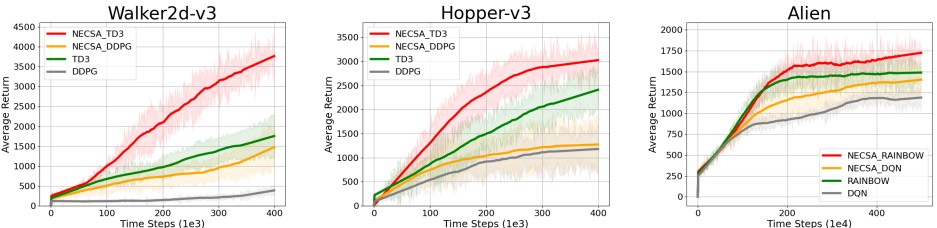

Figure 11: Comparison of performance between building NECSA on different backbones.

## A.4 PERFORMANCE OF APPLYING NECSA TO DMCONTROL TASKS

Figure 12 shows the performance of NECSA on DMControl (Tunyasuvunakool et al., 2020) tasks. In this evaluation, we build NECSA on DrQ-v2 (Yarats et al., 2021), an approach to solve continuous tasks with image-based states. NECSA outperforms DrQ-v2 on Walker-Walk and Hopper-Stand, and achieves the same performance on Cartpole-Balance. The results show that (1) NECSA improves the sample efficiency of DrQ-v2 and (2) NECSA is general and effective for DRL tasks with image-based states and continuous action spaces.

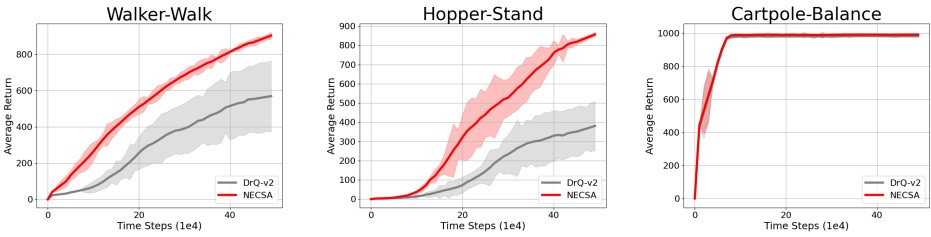

Figure 12: Performance of applying NECSA to DMControl tasks.

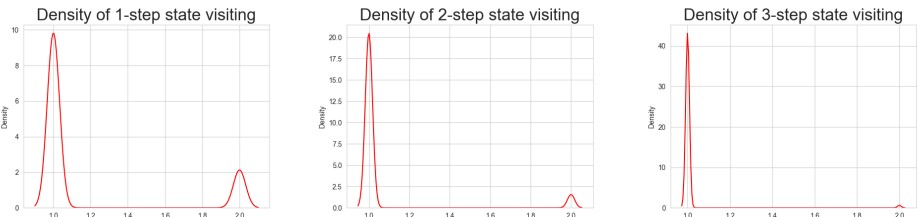

Figure 13: Density of abstract state visiting times on Walker2d-v3.

## A.5 ANALYSIS OF STATE VISITING DENSITY

We plot the density of state visiting times in Figure 13. Take Walker2d-v3 as an example. The 1-step NECSA generated 198,613 abstract states, the 2-step NECSA generated 282,717 abstract patterns, and the 3-step NECSA generated 377,629 abstract patterns. In 1-step setting, 163,273 abstract patterns were visited by only one time, and 35,340 were visited more than one time. In 3-step setting, 372,253 abstract patterns were visited one time, and 5,376 of them were visited more than

one time. All 5,376 multi-times visited patterns were visited 27,747 times, more than the one-time visited pattern.

The density plot shows that the number of 1-time visits increases in multi-step statr abstraction. However, the performance in Figure 5 shows that multi-step analysis can still achieve the best sample efficiency in our settings. Therefore, we assume that the multi-times visited patterns played an essential role in enhancing the sample efficiency. Neverthleass, we assume that there exists better choice of the abstraction steps (*e.g.*, 5-step, 10-step). Due to the abstraction state aprsity issue, we might need a novel state encoder instead of grid-based abstraction. We leave the above assumption as future work.

