# OpenReview forum: "Neural Episodic Control with State Abstraction"
_ICLR.cc/2023/Conference — ICLR 2023 notable top 25%_

### Official Review · Reviewer_QTYi · 2022-10-23

**Confidence:** 3
**Correctness:** 3
**Technical Novelty And Significance:** 3
**Empirical Novelty And Significance:** 3
**Recommendation:** 8

**Clarity, Quality, Novelty And Reproducibility:**

**Clarity**

As mentioned above, although the paper is well-organized and the presentation does not have major flaws, I believe it could still be improved for clarity.

**Quality**

I believe that the quality of the paper, as a whole, is good. The paper is well organized and the presentation strikes me as technically sound.

**Novelty**

Although I am not too familiar with episodic control approaches, as far as I know, the proposed approach is novel. However, I would like to understand better the relationship between the proposed approach and reward shaping.

**Reproducibility**

I believe that the paper is reproducible.

**Strength And Weaknesses:**

**Strong points:**

The problem addressed in the paper - sample efficiency in reinforcement learning - is a relevant problem in reinforcement learning. The proposed approach follows several recent works on the topic of episodic control and -- to my knowledge -- is novel. The paper also strikes me as technically sound.

**Weak points:**

Overall, the writing of the paper could be improved for clarity. Mainly, I felt that Section 4 often went on to discuss some element that was not yet introduced, or was introduced only briefly, which made the presentation at times difficult to follow.

Additionally, the motivation for several choices in the proposed approach is not always evident. Examples include considering multi-step patterns, using scores, rather than returns, or using the scores as intrinsic rewards, rather than improving the policy evaluation.

**Questions for authors**
* The proposed approach strikes me as very similar to _reward shaping_. Specifically, the proposed intrinsic reward mechanism reminds me of potential-based reward shaping, where the scores actually play the role of potential function (and actually provide estimates of the value function). Is this the case? If so, can this relation to reward shaping explain the improved performance observed by the method?

* Although the appendix provides some added discussion regarding the impact of discretization, how much does the discretization impact the performance of the algorithm? It seems to me that, if the discretization is too coarse, the performance of the proposed method will be significantly impacted. On the other hand, for large-dimensional problems, it seems to me that a fine discretization may render the proposed approach infeasible, due to the curse of dimensionality. In Section 5 the paper mentions the use of random projections in the Humanoid-v3 scenario, but I believe that more discussion on this aspect is necessary.

**Summary Of The Paper:**

The paper addresses the problem of sample efficiency in RL by proposing a new episodic control approach. Episodic control consists of maintaining a table-like structure that can easily be accessed to return values associated with states/state-action pairs. The paper proposes NECSA, which introduces several novelties with respect to existing episodic control methods:

1. It discretizes the state space and uses (1-, 2- and 3-state) tuples of discretized states as indices to the episodic memory (an _abstract state_);

2. Each abstract state/state-action pair is mapped to a _score_, which is the average return of all episodes in which such abstract state appears;

3. Scores are used as intrinsic rewards that complement the environment's original reward.

The proposed approach is shown empirically to outperform competing approaches in several continuous control benchmark problems.

**Summary Of The Review:**

The paper addresses a relevant problem and proposes an interesting and novel approach with good empirical results. The relation with reward shaping could be better discussed, as well as the impact of discretization on performance.

---

> ### Author Response · Authors · 2022-11-13
> **Response to Reviewer QTYi (Part1)**
>
> **1. Overall, the writing of the paper could be improved for clarity. Mainly, I felt that Section 4 often went on to discuss some element that was not yet introduced, or was introduced only briefly, which made the presentation at times difficult to follow.**
>
> Thanks for your advice. We would further conduct a thorough polish of the paper, including Section 4, and take your suggestions to refine the statement.
>
> **2. Additionally, the motivation for several choices in the proposed approach is not always evident. Examples include considering multi-step patterns, using scores, rather than returns, or using the scores as intrinsic rewards, rather than improving the policy evaluation.**
>
> Thanks for your questions. We explain the motivation for each part as follows and will polish the paper in the revised version, including the updated contents:
>
> - (Multi-step patterns) In the multi-step analysis, NECSA aims to abstract topological state transitions. In this way, NECSA can identify those behaviors that may negatively affect the upcoming states. The topological relationships between the state transitions improve policy performance [3][4]. The experimental results (shown in Figure 5) proved the effectiveness of the multi-step analysis.
>
> - (Scores vs. Returns) Based on scores, we measure the pattern for forward returns and the entire trajectory. The scores contain more historical information than the returns. Moreover, as stated in Section 1, there is no standard definition of a multi-step pattern. Therefore, the score could potentially be a more comprehensive and effective measurement of similar patterns instead of every single one. We have compared the performance with scores in the ablation study. The results prove that using scores achieves higher sample efficiency.
>
> - (using intrinsic rewards) The scores can also be used for policy evaluation in reinforcement learning since we compute the scores based on episodic returns. Although the pattern is an abstraction of the concrete states, the score represents the evaluation of the policy on the subjective state (or transition) space. Therefore, using scores as intrinsic rewards can also be helpful for improving policy evaluation.
>
>
> **3. The proposed approach is very similar to reward shaping. Specifically, the proposed intrinsic reward mechanism reminds me of potential-based reward shaping, where the scores actually play the role of potential function (and actually provide estimates of the value function). Is this the case? If so, can this relation to reward shaping explain the improved performance observed by the method?**
>
> Thanks for the questions. Our method is mostly inspired by reward-shaping-based methods [5,6,7,8,9]. The use of intrinsic rewards can explain improved performance. The core of our method is to measure the abstract patterns by scores and make the agent generalize the high-score experiences. Reward shaping helps to incorporate domain knowledge into policy optimization. Therefore, it is natural to reshape the reward with a score-based intrinsic reward. The experimental results prove that using reward shaping improves performance. We will update the relevant statement and citations in the paper to highlight the advantages of using reward shaping.

---

> > ### Comment · Reviewer_QTYi · 2022-11-17
> > **Response to author's rebuttal**
> >
> > I thank the authors for the clarifications on the points I raised. I am happy with the author's responses.

---

> > > ### Author Response · Authors · 2022-11-19
> > > **Response to Reviewer QTYi**
> > >
> > > Thanks for the recognition and advice to NECSA. We will continuously revise the paper and improve the quality of NECSA.

---

> ### Author Response · Authors · 2022-11-13
> **Response to Reviewer QTYi (Part2 & Reference)**
>
> **4. Although the appendix provides some added discussion regarding the impact of discretization, how much does the discretization impact the performance of the algorithm? It seems to me that, if the discretization is too coarse, the performance of the proposed method will be significantly impacted. On the other hand, for large-dimensional problems, it seems to me that a fine discretization may render the proposed approach infeasible, due to the curse of dimensionality. In Section 5 the paper mentions the use of random projections in the Humanoid-v3 scenario, but I believe that more discussion on this aspect is necessary.**
>
> Thanks for the questions and advice. We give the explanations as follows:
>
> - (How much does the grid number affect the performance?) In the experiments, we found that a smaller grid number (e.g., 2) enables a more efficient state abstraction but potentially worse performance since too many concrete states (-action) pairs are located in the same abstract pattern. A proper grid number (e.g., 10) would significantly improve the performance, but a much greater grid number makes the abstraction lose generalization and brings the sparsity issue. We have tuned the hyperparameter on Walker2d-v3 and Hopper-v3, finding that the grid set the number to [8,10] is among the best. We inherit this setting to other tasks. We find that too sparse and too dense of a grid will potentially affect the performance, while the effect is small in a reasonable scope (as suggested in the code package).
>
> - (How we perform the gaussian random projection on the high-dimensional tasks) We use a Gaussian random matrix to reduce the dimension of state (-action) vectors. When creating the matrix, we ensure that the values of the matrix elements follow the Gaussian distribution. By multiplying the state (-action) vector (e.g., 1x376) with the Gaussian random matrix (376x24), we can obtain a smaller state (-action) representation vector (1x24). The Gaussian random matrix is initialized at the beginning of the training. Then, it remains the same during the training. All the state (-action) vectors are transformed into a smaller one by a common Gaussian random matrix. During the initialization stage of the Gaussian random matrix, we randomly initialized the value of elements to [0,0.1]. The related experimental results on Humanoid-v3 and HalfCheetah-v3 demonstrate that using a Gaussian random matrix to reduce the size of state (-action) vectors is effective.
>
> - We will further polish the statement in the paper.
>
> **Reference**
>
> [1] Marek Grzes and Daniel Kudenko. Multigrid reinforcement learning with reward shaping. International Conference on Artificial Neural Networks, pp. 357–366. Springer, 2008.
>
> [2] Charles W. Anderson and Stewart Crawford-Hines. Multigrid q-learning. 1994.
>
> [3] Yin, Zhao-Heng, and Wu-Jun Li. "TOMA: Topological Map Abstraction for Reinforcement Learning." arXiv preprint arXiv:2005.06061 (2020).
>
> [4] Kuznetsov, Igor, and Andrey Filchenkov. "Solving Continuous Control with Episodic Memory." arXiv preprint arXiv:2106.08832 (2021).
>
> [5] Andrew Y Ng, Daishi Harada, and Stuart Russell. Policy invariance under reward transformations: Theory and application to reward shaping. In Proceedings of the 16th International Conference on Machine Learning (ICML’99), pages 278–287, 1999.
>
> [6] Anna Harutyunyan, Sam Devlin, Peter Vrancx, and Ann Nowe. Expressing arbitrary reward functions as potential-based advice. In Proceedings of the 29th AAAI Conference on Artificial Intelligence (AAAI’15), pages 2652–2658, 2015.
>
> [7] Adam Laud and Gerald DeJong. The influence of reward on the speed of reinforcement learning: An analysis of shaping. In Proceedings of the 20th International Conference on Machine Learning (ICML’03), pages 440–447, 2003.
>
> [8] Hu, Yujing, et al. Learning to utilize shaping rewards: A new approach of reward shaping. Advances in Neural Information Processing Systems 33 (2020): 15931-15941.
>
> [9] Burda, Yuri, et al. "Exploration by random network distillation." arXiv preprint arXiv:1810.12894 (2018).

---

### Official Review · Reviewer_4xFQ · 2022-10-24

**Confidence:** 4
**Correctness:** 4
**Technical Novelty And Significance:** 4
**Empirical Novelty And Significance:** 3
**Recommendation:** 6

**Clarity, Quality, Novelty And Reproducibility:**

I like this paper in general. The motivation of this paper is clear and make senses to me. The clarity is good. Related work is fully survey. The authors discuss about both limitation and advantages of NECSA.

The results are significant, and the writing quality is good. The proposed approach is novel. The authors report most of hyper-paremeter settings and open-source code. I believe this approach can be reproduced.


Typos:
1. Section 2.1, concrete action space -> continuous action space
2. Caption in Figure 2, s_13 -> s_{13}

**Strength And Weaknesses:**

Strength:
+ The proposed approach is technically sound, novel, scalable and works well across a wide range of continuous control tasks compared with state-of-the-art RL baselines.
+ The idea of building a more comprehensive episodic memory is interesting and differs from existing episodic-memory based approaches, which may benefit the research in related communities.
+ The authors demonstrate a full experimental analysis for the approach, revealing the main contributions, which makes the whole methods more convincing.

Weaknesses:
- NECSA seems a little bit sensitive to hyper-parameter choices. For instance, the optimal value of \epsilon differs across tasks (e.g., 0.2 for Walker2d-v3, 0.1 for Hopper-v3).
- The method so far has been limited to small steps of transition patterns (m=3).
- NECSA was not evaluated on discrete domains such as Atari games (which is the most popular benchmark for most of episodic control based work).

**Summary Of The Paper:**

This paper proposes a novel neural episodic-control based approach with state abstraction, named NECSA. The authors motivate their methods by utilizing latent information from historical behaviors, which has been overlooked by prior work. Specifically, this paper proposes a more comprehensive episodic memory, which consists of state abstraction, multi-step analysis of pattern transitions as well as a novel state measurement metric (reward confidence scores). The proposed algorithm is simple and clear. First, NECSA adopts grid-based state abstraction, tokenizing each state dimension, which achieves efficient state clustering. Second, it extracts patter transitions by sliding a N-step window on each trace and updates reward confidence scores for each patter transition into episodic memory. Third, the episodic memory is used to reform intrinsic reward during policy optimization, to encourage the policy to visit states that lead to highly-rewarded trajectories. Experiments on MuJoCo continuous control tasks demonstrate that the proposed approach works well and outperforms state-of-the-art parameteric methods and non-parameterized episodic-control based baselines. Extensive experiments are conducted for ablation study and hyper-parameter sensitivity.

**Summary Of The Review:**

The paper proposes a novel algorithm named NECSA, which outperforms state-of-the-art episodic-control based methods. The authors conduct extensive experiments to demonstrate the effectiveness of their approach. I believe the overall method can benefit future research in RL community. It would be better to evaluate NECSA on a wide range of Atari games to see whether it is scalable in discrete domains.

---

> ### Author Response · Authors · 2022-11-13
> **Response to Reviewer 4xFQ (Part1)**
>
> **1. NECSA seems a little bit sensitive to hyper-parameter choices. For instance, the optimal value of $\epsilon$ differs across tasks (e.g., 0.2 for Walker2d-v3, 0.1 for Hopper-v3).**
>
>
> Thanks a lot for your questions. We answer these questions as follows:
>
> - (Why does the parameter $\epsilon$ establish a sensitive impact?) The parameter $\epsilon$ determines the proportion of score-based intrinsic and environmental rewards, which is important to policy optimization. The key idea of NECSA is to combine intrinsic rewards with policy optimization. As shown in Equation 4, a greater $\epsilon$ might make the intrinsic rewards cover up the environmental rewards, which might harm the policy performance. However, a smaller $\epsilon$ could decrease the potential influence of intrinsic rewards, thus making NECSA less effective. We follow guidance to select the value of $\epsilon$ in the experiments.
>
> - (how do we determine the value of $\epsilon$?) We provide some guidances to configure the parameter:
>
> 	-  The environmental rewards and the intrinsic rewards should be in the same magnitude of value as those obtained in previous works [1][2], since we follow the same settings and instructions of their original papers.
> 	- The intrinsic rewards should be less than the environmental rewards, the purpose of which is to avoid a significant shift of the optimal policy;
> 	- Most MuJoCo tasks select 0.1 or 0.2 as the value of $\epsilon$. The experimental results demonstrate that this hyperparameter can be commonly used (and shown to be useful) in different tasks.
>
> - We agree that $\epsilon$ is important for policy optimization and its ultimate performance. We reported the details, suggested the proper selection scope, and opened the best parameter settings in our build. We hope the above information could be helpful for the usage of NECSA, and we also believe that more advanced $\epsilon$ selection methods could be important and we leave it as future work.
>
>
> **2. The method so far has been limited to small steps of transition patterns (m=3).**
>
> | Task Name | 1-step| 2-step| 3-step| 4-step| 5-step| 6-step|
> | :----: |:----:| :----:| :----:|:----:| :----:|:----:|
> | Walker2d-v3 | 2334.84 $\pm$ 174.01|2666.12 $\pm$ 352.87|**3768.08 $\pm$ 270.54** |2597.33 $\pm$ 267.37| 2021.80 $\pm$ 335.19 |2308.49 $\pm$ 168.56|
>
> Thanks for your questions. Currently, the multi-step analysis is 3-steps. The environmental results up to the 2-step analysis demonstrate that (1) 3-step analysis is capable of reaching or even outperforming the state-of-the-art performance, and (2) exploring the multi-step transition is helpful. We conducted experiments on Walker2d-v3 with different m-step analyses (m=1, 2, 3, 4, 5, 6). The results can be found in an anonymous link [multi-steps](https://anonymous.4open.science/r/NECSA-ICLR-2023/multi-steps.png) (please kindly click the link to view). Specifically, NECSA_3-step achieves the highest average returns, 3768.08, much higher than the longer-step settings. The results show that 3-step pattern analysis could be the optimal choice (in MuJoCo tasks). However, the longer-step analysis also brings issues. Regarding the problem of sparsity, the number of multi-step abstract patterns significantly decreases as the step increases. This may affect the performance of NECSA. Besides, we are considering a more generalized episodic memory to enable more complex and longer pattern analysis (e.g., encoding the multiple states by an LSTM network), and we leave this as future work.

---

> ### Author Response · Authors · 2022-11-13
> **Response to Reviewer 4xFQ (Part2 & Reference)**
>
> **3. NECSA was not evaluated on discrete domains such as Atari games (which is the most popular benchmark for most of episodic control based work).**
>
> | Task Name | GEM    | NECSA_1-step| NECSA_2-step| NECSA_3-step|
> |  :----:  |    :----:   |    :----:  |    :----: |   :----:  |
> | Alien | 1284.55 $\pm$ 50.72 | 1441.85 $\pm$ 71.03 | 1480.58 $\pm$ 129.83 | **1731.14 $\pm$ 78.31** |
>
> Thanks for your questions and suggestions. NECSA is general to the discrete domains, e.g., including Atari games. NECSA performs an abstraction on the inputs of the policy network using a grid-based method [8,9]. In MuJoCo domains, the inputs are concrete vectors that are suitable for grid-based abstraction. However, in discrete action spaces, e.g., Atari, the policy network takes the high-dimensional images as input. Therefore, applying the grid-based abstraction to these images might be challenging. To address this problem, we instead use hidden outputs of the policy network for abstraction, i.e., the hidden outputs are the features of the images [10].
>
> Based on this, we also conducted experiments on an Atari game, Alien. The obtained results can be found in [Atari_Alien](https://anonymous.4open.science/r/NECSA-ICLR-2023/Atari_Alien.png) (please kindly click the link to view). In Alien, NECSA_3-step achieves the highest average returns, 1731.14 over five seeds, and outperforms GEM with 1284.55 average returns.   NECSA_3-step also outperforms the 1-step and 2-step settings. The result demonstrates that NECSA can be applied to discrete domains (e.g., Atari games) and outperforms the baseline method. Moreover, the main conclusions based on Alien is the consistent as those obtained on MuJoCo tasks: a multi-step analysis benefits policy optimization.
>
> **4. The typo issues the reviewer motioned above.**
>
> Thanks for the suggestions. We will thoroughly check the paper presentation and polish it further in the revised version.
>
> **Reference**
>
> [1] Burda, Yuri, et al. "Exploration by random network distillation." arXiv preprint arXiv:1810.12894 (2018).
>
> [2] Devlin, Sam Michael, and Daniel Kudenko. "Dynamic potential-based reward shaping." Proceedings of the 11th international conference on autonomous agents and multiagent systems. IFAAMAS, 2012.
>
> [3] Marek Grzes and Daniel Kudenko. Multigrid reinforcement learning with reward shaping. International Conference on Artificial Neural Networks, pp. 357–366. Springer, 2008.
>
> [4] Charles W. Anderson and Stewart Crawford-Hines. Multigrid q-learning. 1994.
>
> [5] Hu, Hao, et al. "Generalizable episodic memory for deep reinforcement learning." arXiv preprint arXiv:2103.06469 (2021).
>
> [6] Alexander Pritzel, Benigno Uria, Sriram Srinivasan, Adrià Puigdomènech Badia, Oriol Vinyals, Demis Hassabis, Daan Wierstra, Charles Blundell Proceedings of the 34th International Conference on Machine Learning, PMLR 70:2827-2836, 2017.
>
> [7] Lin Zichuan, Zhao Tianqi, Yang Guangwen, Lintao, Zhang. (2018). Episodic Memory Deep Q-Networks. 2433-2439. 10.24963/ijcai.2018/337.
>
> [8] Hansen, Steven Stenberg et al. “Fast deep reinforcement learning using online adjustments from the past.” ArXiv abs/1810.08163 (2018).

---

> ### Author Response · Authors · 2022-11-30
> **Response to Reviewer 4xFQ: Experiments on Atari and DMControl tasks (Part1)**
>
> ## Suggestions about more experiments to demonstrate the effectiveness of NECSA in image-based input environments
>
> We follow the suggestion and have performed more comprehensive evaluation of NECSA on 6 Atari games and 3 DMControl tasks with image-based states. The results also demonstrate that NECSA is general and effective for tasks that take images as states. We report the results and the implementation details of NECSA as follows.
>
> We perform grid-based abstraction on hidden outputs instead of states or state-action pairs on image-states tasks. The reason is that abstracting images directly can be inefficient. Moreover, pixels-based image vectors consist of multiple channels (e.g., RGB images), which can be unsuitable for abstraction using grid-based methods. Therefore, we use the hidden outputs for abstraction, considering that (1) The hidden outputs are natural features of images but relatively smaller than the raw images, and (2) hidden outputs reflect the states of actions. As discussed in the paper (Appendix A.3), abstracting the state-action pairs achieves higher sample efficiency since they contain more information for modelling the policy than just focusing on states. Incorporating hidden outputs with episodic control is also effective in EVA [1].
>
> In particular, we take the hidden outputs after the convolutional neural network layers of the policy network to represent the image states. To reduce the size of the hidden outputs and to improve the computation efficiency of NECSA, we adopt a sigmoid activation layer after the hidden outputs to limit the value of the feature vectors to [0,1] (that are the lower and upper bounds for grid-based abstraction). Our next step is to reduce the dimension of the feature vectors to 24 using Gaussian random projection. In the end, we process the feature vectors through a 5-grid abstraction. Note that the related hyperparameters adopted in our evaluation are the same as those used in MuJoCo tasks.
>
> ## Atari Games
>
> We follow the suggestion and further conduct NECSA experiments on 6 Atari games. The details of the experimental results can be found in the following links. Note that all tasks are unified to the "NoFrameskip-v4" version. In the experiments, to be comprehensive, we compared NECSA to two state-of-the-art episodic control methods EVA [1] and GEM [2], and two baseline methods DQN [3] and Rainbow [4]. All experimental results were obtained on five random seeds in 5 million training steps. Besides, all baseline methods share the same network architecture and hyperparameter settings, which can be found in the anonymous program repository [link](https://anonymous.4open.science/r/NECSA-ICLR-2023/README.md). We also report the average returns of each method after the respective training steps in the experimental results. The results demonstrate and confirm that NECSA outperforms the baselines and the state-of-the-art episodic control methods across all the settings. This confirms that NECSA can be general and effective on image-states tasks as well.
>
>
> | Task Name | DQN| RAINBOW| EVA| GEM| NECSA|
> | :----: |    :----:   |  :----:| :----:| :----:|:----:|
> |[Alien](https://anonymous.4open.science/r/NECSA-ICLR-2023/rebuttal/Alien.png)|1188.73 $\pm$ 58.37|1488.84 $\pm$ 114.93|1432.22 $\pm$ 109.17|1030.67 $\pm$ 48.17|**1725.56$\pm$ 83.03**|
> |[Breakout](https://anonymous.4open.science/r/NECSA-ICLR-2023/rebuttal/Breakout.png)|61.26 $\pm$ 4.01|385.07 $\pm$ 12.33|381.29 $\pm$ 15.63|220.73 $\pm$ 54.09|**412.30 $\pm$ 9.94**|
> |[Enduro](https://anonymous.4open.science/r/NECSA-ICLR-2023/rebuttal/Enduro.png)|748.45 $\pm$ 49.08|824.62$\pm$ 45.77|733.50$\pm$ 24.98|729.60 $\pm$ 49.87|**1147.48 $\pm$ 29.26**|
> |[Qbert](https://anonymous.4open.science/r/NECSA-ICLR-2023/rebuttal/Qbert.png)|9670.13 $\pm$ 464.31|10817.71 $\pm$ 656.88|9617.25 $\pm$ 464.31|9643.69 $\pm$ 492.48|**12659.28 $\pm$ 471.96** |
> | [MsPacman](https://anonymous.4open.science/r/NECSA-ICLR-2023/rebuttal/MsPacman.png)|1997.72 $\pm$ 120.59|2075.16 $\pm$ 106.50|1991.82 $\pm$ 78.71|1990.81 $\pm$ 134.34|**2205.28 $\pm$ 81.38** |
> | [SpaceInvaders](https://anonymous.4open.science/r/NECSA-ICLR-2023/rebuttal/SpaceInvaders.png)|629.85 $\pm$ 42.62|810.12 $\pm$ 49.93|799.06 $\pm$ 26.32|686.25 $\pm$ 53.36| **951.62 $\pm$ 28.03** |

---

> ### Author Response · Authors · 2022-11-30
> **Response to Reviewer 4xFQ: Experiments on Atari and DMControl tasks (Part2)**
>
> Furthermore, we also conducted the ablation studies and hyperparameter experiments on NECSA in the following settings:
> - Ablation Study:
> 	- We compare the performance of NECSA on different multi-step analyses (1, 2, and 3 steps). The results in this [link](https://anonymous.4open.science/r/NECSA-ICLR-2023/rebuttal/Alien-steps.png) show that 3-steps is the most effective setting.
> 	- We compare the performance of NECSA using scores and average Q-values for abstract pattern measurements. The results in this [link](https://anonymous.4open.science/r/NECSA-ICLR-2023/rebuttal/Alien-scores.png) show that using scores as measurement can significantly improve the sample efficiency.
>
> - Hyperparameters:
> 	- We compare the performance of NECSA on abstracting the states and hidden outputs. To be specific, we use a fixed policy network to extract the static features of the image inputs as states, and the real-time policy network extracts the dynamic features as state-action representations. The result in this [link](https://anonymous.4open.science/r/NECSA-ICLR-2023/rebuttal/Alien-state.png) shows that using hidden outputs achieves higher sample efficiency.
> 	- We compare the performance of NECSA on different grid numbers for abstraction. The results in [link](https://anonymous.4open.science/r/NECSA-ICLR-2023/rebuttal/Alien-epsilon.png) show that five grids are the most effective choice for abstraction.
> 	- We compare the performance of NECSA with different values of epsilon. The results in this [link](https://anonymous.4open.science/r/NECSA-ICLR-2023/rebuttal/Alien-grid.png) show that 0.1 is the most suitable epsilon value for reward shaping.
> 	- NECSA is built on Rainbow for Atari tasks. We have also conducted evaluation on the performance of NECSA with different backbone algorithms. The results in [link](https://anonymous.4open.science/r/NECSA-ICLR-2023/rebuttal/Alien-backbone.png) demonstrate that (1) NECSA can achieve significant improvements in sample efficiency on different backbone algorithms and (2) the selection of backbone algorithms can affect the performance of NECSA.
>
> - The above conclusions on Atari tasks are consistent as NECSA for MuJoCo tasks. We use the same hyperparameter settings for all the Atari and DMControl tasks.
>
> Overall, we conducted large-scale and relatively comprehensive experiments on 6 Atari games to demonstrate that NECSA is general and effective for most DRL tasks. Please refer to our [program] (https://anonymous.4open.science/r/NECSA-ICLR-2023/README.md) for more details.
>
> ## DMControl Tasks with image-based states
> We tested NECSA on DMControl tasks with image-based states. We compare NECSA to DrQ-v2 [5]. DrQ-v2 is the state-of-the-art method for solving continuous control tasks with image-based states. Note that existing episodic control methods (e.g., GEM) are not applicable to continuous control tasks with image-based states, forcing us unable to compare with them. We select three continuous control tasks: Walker-Walk, Hopper-Stand, and Cartpole-Balance. All the experiments run in 500,000 training steps on five random seeds. We strictly follow the hyperparameter settings and network architecture in Appendix B of DrQ-v2 [5] to enable the fair comparison.  The implementation repository is in the anonymous [link](https://anonymous.4open.science/r/drqv2_necsa/README.md). The overall results are reported as follows:
>
> | Task Name | DrQ-v2| NECSA|
> |:----:|:----:|:----:|
> |[Walker-Walk](https://anonymous.4open.science/r/drqv2_necsa/images/Walker-Walk.png)|569.28 $\pm$ 102.80|**903.36$\pm$ 28.77**|
> |[Hopper-Stand](https://anonymous.4open.science/r/drqv2_necsa/images/Hopper-Stand.png)|380.51 $\pm$ 66.79|**856.59$\pm$ 61.62**|
> |[Cartpole-Balance](https://anonymous.4open.science/r/drqv2_necsa/images/Cartpole-Balance.png)|983.29 $\pm$ 10.66|**992.31 $\pm$ 10.58**|
>
> Our results reveal that NECSA can significantly outperform DrQ-v2 on DMControl tasks. In terms of the results on MuJoCo (in the paper), Atari, and DMControl domains, we could conclude that NECSA is effective on (1) concrete states and pixel-based states, (2) continuous and discrete action spaces and (3) a mixture of them. We believe that NECSA can be applied to most DRL tasks and improve various DRL algorithms.
>
> ## Reference
> [1] Hu, Hao, et al. "Generalizable episodic memory for deep reinforcement learning." _arXiv preprint arXiv:2103.06469_ (2021).
>
> [2] Hansen, Steven, et al. "Fast deep reinforcement learning using online adjustments from the past." _Advances in Neural Information Processing Systems_ 31 (2018).
>
> [3] Mnih, Volodymyr, et al. "Playing atari with deep reinforcement learning." _arXiv preprint arXiv:1312.5602_ (2013).
>
> [4] Hessel, Matteo, et al. "Rainbow: Combining improvements in deep reinforcement learning." _Thirty-second AAAI conference on artificial intelligence_. 2018.
>
> [5] Yarats, Denis, et al. "Mastering visual continuous control: Improved data-augmented reinforcement learning." _arXiv preprint arXiv:2107.09645_ (2021).

---

### Official Review · Reviewer_D6zD · 2022-10-28

**Confidence:** 4
**Correctness:** 4
**Technical Novelty And Significance:** 3
**Empirical Novelty And Significance:** 2
**Recommendation:** 8

**Clarity, Quality, Novelty And Reproducibility:**

Clarity: very clear, notations could be improved

Quality: good

Novelty: the contributions (especially the grid projection) are sound, simple and interesting but all together not extremely novel

Reproducibility: the data and code are claimed to be provided but I was not able to retrieve them at the given link


**Strength And Weaknesses:**

### Strengths

The paper is well-written and easy to follow.

The contributions are clearly stated and suitable experiments are proposed to validate the paper claims.

Both the grid abstraction and multi-step analysis are interesting ideas that seem to work well in the explored context.

The method performs well on a variety of Mujoco tasks.

### Weaknesses

The notations and overall math of the paper could really be improved (for instance using \pi(\cdot | s), s_{t + 1}, r(s, a), etc), though I do not think this is a major issue for the comprehension in the current state of the paper.

The code and data (unless I am wrong) are not available at the specified URL.

The grid definition is ad-hoc to state spaces with a relatively limited number of components (though it works for Humanoid). Do you have ideas to extend it to synthetic/natural images for instance? This would make the method applicable to other environments (e.g. DMControl, Atari, etc).

I would advocate for a short description of how the gaussian random projections work in 4.1. This is an important component of the method whose details are currently missing.

I would also like the paper to have a small explanation of how actions are handled when using state-action pairs for grid abstraction.

Did you try different intrinsic rewards? While it appears natural to reward state-action pairs proportionally to how much their average return is superior to the average over all state-action-pairs, it might be interesting to use an advantage approximation as reward instead: the approximate advantage could be $\c(s_t, a_t) - \mathbb{E}_{a \sim \pi}[c(s_t, a)]$ for instance. This would connect the method with Advantage Learning methods (Baird, 1995).

In Table 1, it is stated that “the best average returns” are reported. Does that mean that only the average return of the best seed is reported? If so, I recommend to report the average over seeds instead (and standard deviation).

The impact of the paper would be greater with DMControl experiments included (but that requires an extension to the grid abstraction).


**Summary Of The Paper:**

The paper introduces a state abstraction method for neural episodic control. States or state-action pairs are mapped to grid coordinates and dimensionality is reduced further by random projections. The method is suitable for state spaces with a limited number of independent dimensions. The method provides better results than the baselines on Mujoco.

**Summary Of The Review:**

The paper is well-written, and makes for an enjoyable read. Contributions are backed by relevant experiments. The method is conceptually simple and elegant.
Though, as I mentioned in my comments, the lack of extension of the grid abstraction to images limits the relevance of the ideas presented, due to the nature of benchmarks in Deep RL. Also, most of the notations could be reworked (see comments).

I am borderline recommending for acceptance since I like the paper and its ideas but I really think that the mentioned extension would really improve its potential impact.

---

> ### Author Response · Authors · 2022-11-13
> **Response to Reviewer D6zD (Part1)**
>
> **1. The notations and overall math of the paper could really be improved (for instance using $\pi(\cdot | s), s_{t + 1}, r(s, a)$, etc), though I do not think this is a major issue for the omprehension in the current state of the paper.**
>
> Thanks for your suggestions. We would further thoroughly polish the paper and improve the notations.
>
> **2. The code and data (unless I am wrong) are not available at the specified URL.**
>
> Thanks for mentioning this. We have submitted the source code as supplementary materials in OpenReview. Besides, we have also added an open-access [anonymous link](https://anonymous.4open.science/r/NECSA-ICLR-2023/README.md) on the website of NECSA. Please kindly check more details on the anonymous link.
>
> **3. The grid definition is ad-hoc to state spaces with a relatively limited number of components (though it works for Humanoid). Do you have ideas to extend it to synthetic/natural images for instance? This would make the method applicable to other environments (e.g. DMControl, Atari, etc).**
>
> | Task Name | GEM    | NECSA_1-step| NECSA_2-step| NECSA_3-step|
> | :----: |    :----:   |  :----:| :----:| :----:|
> | Alien | 1284.55 $\pm$ 50.72 | 1441.85 $\pm$ 71.03 | 1480.58 $\pm$ 129.83 | **1731.14 $\pm$ 78.31** |
> | Walker-Walk | 219.70 $\pm$ 18.16 | 750.94 $\pm$ 32.94 | 818.93 $\pm$ 33.43 | **880.74 $\pm$ 22.10** |
>
>  NECSA is a general approach that can be used in both continuous and discrete action spaces.
> - The difference is that concrete inputs (e.g., with float numbers) could be directly handled with grid-based abstraction, while image-based inputs (e.g., with pixels) could not. To process image-based inputs, we need to perform another abstraction that first converts the image-based inputs to concrete features. Then, NECSA can be adopted. Specifically, we could take the hidden outputs of the policy network as the input of NECSA. The hidden outputs can be considered as an abstraction of the image inputs.
>
> - We have also conducted additional experiments on Atari games and DMControl to demonstrate the usefulness and potential of NECSA in the discrete domain. For example, Alien requires the agent to fight in a maze-like space and defeat the enemies. Walker-walk requires the agent to walk robustly. In Alien, all the settings of NECSA achieve higher returns than the baseline approach, GEM. NECSA_3-step achieves the best average returns, 1731.14 over five seeds. In Walker-Walk, all the settings of NECSA obviously outperform GEM that only achieves 219.70 returns. NECSA_3-step achieves 880.74 with better performance. The results on Atari Alien and DMControl demonstrate that NECSA outperforms the baseline approach, and the multi-step approach achieves higher sample efficiency than the 1-step setting. Both results on Atari and DMControl further confirm that NECSA is effective on various DRL tasks. The plots of the detailed results are in anonymous links here, [Atari_Alien](https://anonymous.4open.science/r/NECSA-ICLR-2023/Atari_Alien.png) and [DMControl](https://anonymous.4open.science/r/NECSA-ICLR-2023/DMControl.png), respectively (please kindly click the link to view the results).
>
> - Thanks for your question and suggestion, and we would add more discussion in the revision.
>
> **4. I would advocate for a short description of how the gaussian random projections work in 4.1. This is an important component of the method whose details are currently missing.**
>
> Thanks for your suggestions. We would polish the paper and add the details of Gaussian random projection in the revised version. In general, the high-dimensional concrete state (-action pairs) would increase the difficulty and complexity of state abstraction. Inspired by [4], we applied Gaussian random projection by a matrix to reduce the dimensions of state (-action) vectors. When creating the matrix, we ensure that the values of the matrix elements follow the Gaussian distribution and that the value scope is [0,0.1]. By multiplying the state (-action) vector (e.g., 1x376) with the Gaussian random matrix (376x24), we can obtain a smaller state (-action) representation vector (1x24). The Gaussian random matrix is initialized at the beginning of the training. Then, it would remain the same during the training. Finally, all the state (-action) vectors are projected to a smaller one by a common Gaussian random matrix. The related experimental results on Humanoid-v3, Ant-v3, and HalfCheetah-v3 (see Figure 4) demonstrate that leveraging a Gaussian random projection to reduce the size of state (-action) vectors is effective.

---

> ### Author Response · Authors · 2022-11-13
> **Response to Reviewer D6zD (Part2)**
>
> **5. I would also like the paper to have a small explanation of how actions are handled when using state-action pairs for grid abstraction.**
>
> Thanks for the question. We report the details and will update the related statement in the revision as suggested. We take a Walker2d-v3 as an example for the explanation. The shape of the state vector is (1x17), and the action vector is (1x6). Then, we combine the state and action vectors into a (1x23) state-action pair. The lower and upper bounds of the state vectors are [-10,10]. The lower and upper bounds of the action vectors are [-1,1] in Walker2d-v3. Finally, for instance, we obtain the fixed lower bounds as (-10, -10, ..., -10, -1, -1, ..., -1). Next, we abstract the state-action pairs by grid-based clustering. Each dimension of a state-action pair is split into equal intervals.
>
> **6. Did you try different intrinsic rewards? While it appears natural to reward state-action pairs proportionally to how much their average return is superior to the average over all state-action-pairs, it might be interesting to use an advantage approximation as reward instead: the approximate advantage could be $c(s_t,a_t)-E_{a \sim\pi}[c(s_t,a)]$ for instance. This would connect the method with Advantage Learning methods (Baird, 1995).**
>
>   | Task Name | NECSA_advantage_1| NECSA_advantage_2| NECSA_advantage_3| NECSA |
> |  :----:     |    :----:   |    :----:  |    :----:  |    :----:  |
> | Walker2d-v3 |3011.16 $\pm$ 235.73|2548.12 $\pm$ 224.95|3231.15 $\pm$ 180.14|**3768.08 $\pm$ 270.54** |
>
> Thanks for your questions and advice. We appreciate this idea. Using an advantage-based method on scores is reasonable and exciting. We have conducted experiments on the advantage-based intrinsic rewards on Walker2d-v3 (the result is in an anonymous link here [score_advantage](https://anonymous.4open.science/r/NECSA-ICLR-2023/score_advantage.png)(please click the link to view). The result shows that (1) using advantage-based intrinsic rewards can significantly improve the sample efficiency against the baselines (compared to the results in Figure 4), e.g., NECSA_advantage_3 achieves 3231.15 average returns, which is higher than the EMAC and GEM; (2) our methods achieve higher sample efficiency than the advantage-based intrinsic rewards, NECSA is 3768.08 and higher than advantage-based settings. The reason is that advantage-based intrinsic rewards are sometimes 0. We found that the occurrence of some abstract patterns is 1. In this case, the score $c(s_t,a_t)=E_{a \sim\pi}[c(s_t, a)]$ since $a=\{a_t\}$. Therefore, the advantage-based intrinsic rewards fail to benefit the policy optimization in the above cases.
>
> __7. In Table 1, it is stated that "the best average returns" are reported.__ __Does that mean that only the average return of the best seed is reported?__ __If so, I recommend to report the average over seeds instead (and standard deviation).__
>
> Thanks for your suggestions. The experimental data (including the best average returns) are all over the runs of multiple random seeds. We would add the standard deviation to the main results table and polish the statement to make it more precise.
>
> **8. The impact of the paper would be greater with DMControl experiments included (but that requires an extension to the grid abstraction).**
>
> Thanks for your advice. NECSA can also be applied to DMControl tasks. To demonstrate the effectiveness of NECSA, we follow your suggestions and conduct the additional experiments on "Walker-Walk", a DMControl suit. The experiment [results](https://anonymous.4open.science/r/NECSA-ICLR-2023/DMControl.png) (please click the link to view the results) confirm the usefulness of NECSA.

---

> ### Author Response · Authors · 2022-11-13
> **Response to Reviewer D6zD (Reference)**
>
> **Reference**
>
> [1] Alexander Pritzel, Benigno Uria, Sriram Srinivasan, Adrià Puigdomènech Badia, Oriol Vinyals, Demis Hassabis, Daan Wierstra, Charles Blundell Proceedings of the 34th International Conference on Machine Learning, PMLR 70:2827-2836, 2017.
>
> [2] Lin Zichuan, Zhao Tianqi, Yang Guangwen, Lintao, Zhang. (2018). Episodic Memory Deep Q-Networks. 2433-2439. 10.24963/ijcai.2018/337.
>
> [3] Hu, Hao, et al. "Generalizable episodic memory for deep reinforcement learning." arXiv preprint arXiv:2103.06469 (2021).
>
> [4] Dasgupta, Sanjoy. "Experiments with random projection." arXiv preprint arXiv:1301.3849 (2013).
>
> [5] Kuznetsov, Igor, and Andrey Filchenkov. "Solving Continuous Control with Episodic Memory." arXiv preprint arXiv:2106.08832 (2021).
>
> [6] Duan, Yan, et al. "Benchmarking deep reinforcement learning for continuous control." International conference on machine learning. PMLR, 2016.
>
> [7] Lillicrap, Timothy P., et al. "Continuous control with deep reinforcement learning." arXiv preprint arXiv:1509.02971 (2015).
>
> [8] Marek Grzes and Daniel Kudenko. Multigrid reinforcement learning with reward shaping. International Conference on Artificial Neural Networks, pp. 357–366. Springer, 2008.
>
> [9] Charles W. Anderson and Stewart Crawford-Hines. Multigrid q-learning. 1994.
>
> [10] Hansen, Steven Stenberg et al. “Fast deep reinforcement learning using online adjustments from the past.” ArXiv abs/1810.08163 (2018): n. pag.

---

> > ### Comment · Reviewer_D6zD · 2022-11-15
> > **Response to authors' response**
> >
> > 1. Good.
> >
> > 2. Thanks, acknowledged.
> >
> > 3. How do you handle the representational drift with the proposed abstraction in Atari? I.e. the policy network will evolve along training, so similar states will be projected onto different grid coordinates. If this works well, I would argue for including results over several Atari games in the paper as this would really increase the impact of the paper.
> >
> > 4. Thanks for the detailed explanation.
> >
> > 5. So, concatenation, which makes sense. This should be added to the paper.
> >
> > 6. I appreciate that authors took the time to test the proposed method.
> >
> > 7. I do not question the fact that several seeds are used. My question is: what do authors mean by best average? Please define it clearly, and consider reporting the average over all seeds.
> >
> > 8. Sorry for not making that clear: I meant DMControl with image input. But additional results are welcome.

---

> > > ### Author Response · Authors · 2022-11-19
> > > **Response to Reviewer D6zD**
> > >
> > > **Suggestions about more experiments to demonstrate the effectiveness of NECSA in image-based input environments**
> > >
> > > Thanks for your suggestion. We will continuously conduct more experiments on image-input DMControl tasks and add the results. In Atari Alien, we have shown that *NECSA* is also effective for the image input-based environment. Due to time and our available computational resource constraint (the discussion and response period will end in a few hours), we cannot show more results on image-based DMControl and various Atari games environments in the fastly coming rebuttal period. However, we promise that we would add these results to our paper (e.g., as supplementary material or on our paper accompanied website) to further increase the significant impact of our contribution **NECSA** as suggested.
> > >
> > > **3. How do you handle the representational drift with the proposed abstraction in Atari? I.e. the policy network will evolve along training, so similar states will be projected onto different grid coordinates. If this works well, I would argue for including results over several Atari games in the paper as this would really increase the impact of the paper.**
> > >
> > > Thanks for your questions and suggestions.
> > >
> > > - (Handle the representational drift)
> > >
> > > 	- Despite having representational drifts, NECSA is effective on Atari Alien. We use the grid-based abstraction to benchmark the decision-making patterns instead of single states.
> > > 	- As stated in Figure 8 from the Appendix, we have compared the difference between abstracting the concrete states and state-action pairs. The results confirm that using state-action pairs for abstraction significantly improves performance. The above results are helpful for explaining the benefits of using hidden outputs for abstraction since the hidden outputs are the reflections from the states to actions.
> > > 	- Incorporating the hidden outputs with the episodic memory is also demonstrated to be effective in Ephemeral Value Adjustment (EVA) [1]. Previous episodic controls store the states, actions, and the related Q-values to benchmark the episodic state values, while EVA uses the hidden outputs instead of the states. In Atari tasks, NECSA is a further upgrade of EVA since we abstract the hidden outputs and conduct a multi-step analysis on the abstractions.
> > >
> > >
> > > - (Including results over several Atari games) We appreciate the advice of applying NECSA on large-scale Atari tasks. We will continuously perform the evaluation on more tasks, and report the results and implementation details in the finalized version.
> > >
> > > **7. I do not question the fact that several seeds are used. My question is: what do authors mean by best average? Please define it clearly, and consider reporting the average over all seeds.**
> > >
> > > Thanks for your question and suggestions.
> > >
> > > - The **best average** refers to the highest mean episodic return over multiple seeds.
> > >
> > > - Each run on a seed will generate a list of episodic returns. Each return is the evaluation of the policy at the current training steps. For instance, the list length would be 1,000 if we evaluate the policy per 5,000 training steps over a total of 5M training steps. Then, we obtain several return lists over multiple seeds. We use these lists to plot the results in Figure 4. The bold line in Figure 4 represents the average returns over the multiple returns, which is also a list. The **best average**  means the highest value on the average return list (i.e., the highest point of the bold line).
> > >
> > > - Also thanks for your suggestion, we will give a more concrete definition and report the average over all the seeds in the paper. Moreover, we could also consider removing the word "best" since it seems to bring some confusion.
> > >
> > > [1] Hansen, Steven & Sprechmann, Pablo & Pritzel, Alexander & Barreto, Andre & Blundell, Charles. (2018). Fast deep reinforcement learning using online adjustments from the past.

---

> > > > ### Comment · Reviewer_D6zD · 2022-11-28
> > > > **Response to authors**
> > > >
> > > > I want to thank the authors for the clarification and additional elements.

---

> > > > > ### Author Response · Authors · 2022-11-30
> > > > > **Response to Reviewer D6zD**
> > > > >
> > > > > We take the suggestions for applying NECSA to DRL tasks with image-based states. We have conducted large-scale experiments and reported the results in the comments. Please kindly refer to the responses for details.
> > > > >
> > > > > We thank Reviewer D6zD for the advice to improve NECSA.

---

> > > > > > ### Comment · Reviewer_D6zD · 2022-12-01
> > > > > > **Response to authors**
> > > > > >
> > > > > > I want to thank the authors for the extensive additional experiments conducted. I encourage the authors to include these results in the main text and adapt the paper claims accordingly. I have updated my score to a clear accept.

---

> > > > > > > ### Author Response · Authors · 2022-12-01
> > > > > > > **Response to Reviewer Q6zD**
> > > > > > >
> > > > > > > We really appreciate the reviewer's constructive and insightful comments and feedback, which is so much helpful for us to enhance the quality of this work.
> > > > > > > We would follow the suggestion and add these obtained results and adapt corresponding claims in the paper, to make the paper more solid.
> > > > > > > Many thanks again for the very helpful comments and feedback.

---

> ### Author Response · Authors · 2022-11-30
> **Response to ReviewerD6zD: Experiments on Atari and DMControl (Part1)**
>
> ## Suggestions about more experiments to demonstrate the effectiveness of NECSA in image-based input environments
>
> We follow the suggestion and have performed more comprehensive evaluation of NECSA on 6 Atari games and 3 DMControl tasks with image-based states. The results also demonstrate that NECSA is general and effective for tasks that take images as states. We report the results and the implementation details of NECSA as follows.
>
> We perform grid-based abstraction on hidden outputs instead of states or state-action pairs on image-states tasks. The reason is that abstracting images directly can be inefficient. Moreover, pixels-based image vectors consist of multiple channels (e.g., RGB images), which can be unsuitable for abstraction using grid-based methods. Therefore, we use the hidden outputs for abstraction, considering that (1) The hidden outputs are natural features of images but relatively smaller than the raw images, and (2) hidden outputs reflect the states of actions. As discussed in the paper (Appendix A.3), abstracting the state-action pairs achieves higher sample efficiency since they contain more information for modelling the policy than just focusing on states. Incorporating hidden outputs with episodic control is also effective in EVA [1].
>
> In particular, we take the hidden outputs after the convolutional neural network layers of the policy network to represent the image states. To reduce the size of the hidden outputs and to improve the computation efficiency of NECSA, we adopt a sigmoid activation layer after the hidden outputs to limit the value of the feature vectors to [0,1] (that are the lower and upper bounds for grid-based abstraction). Our next step is to reduce the dimension of the feature vectors to 24 using Gaussian random projection. In the end, we process the feature vectors through a 5-grid abstraction. Note that the related hyperparameters adopted in our evaluation are the same as those used in MuJoCo tasks.
>
> ## Atari Games
>
> We follow the suggestion and further conduct NECSA experiments on 6 Atari games. The details of the experimental results can be found in the following links. Note that all tasks are unified to the "NoFrameskip-v4" version. In the experiments, to be comprehensive, we compared NECSA to two state-of-the-art episodic control methods EVA [1] and GEM [2], and two baseline methods DQN [3] and Rainbow [4]. All experimental results were obtained on five random seeds in 5 million training steps. Besides, all baseline methods share the same network architecture and hyperparameter settings, which can be found in the anonymous program repository [link](https://anonymous.4open.science/r/NECSA-ICLR-2023/README.md). We also report the average returns of each method after the respective training steps in the experimental results. The results demonstrate and confirm that NECSA outperforms the baselines and the state-of-the-art episodic control methods across all the settings. This confirms that NECSA can be general and effective on image-states tasks as well.
>
>
> | Task Name | DQN| RAINBOW| EVA| GEM| NECSA|
> | :----: |    :----:   |  :----:| :----:| :----:|:----:|
> |[Alien](https://anonymous.4open.science/r/NECSA-ICLR-2023/rebuttal/Alien.png)|1188.73 $\pm$ 58.37|1488.84 $\pm$ 114.93|1432.22 $\pm$ 109.17|1030.67 $\pm$ 48.17|**1725.56$\pm$ 83.03**|
> |[Breakout](https://anonymous.4open.science/r/NECSA-ICLR-2023/rebuttal/Breakout.png)|61.26 $\pm$ 4.01|385.07 $\pm$ 12.33|381.29 $\pm$ 15.63|220.73 $\pm$ 54.09|**412.30 $\pm$ 9.94**|
> |[Enduro](https://anonymous.4open.science/r/NECSA-ICLR-2023/rebuttal/Enduro.png)|748.45 $\pm$ 49.08|824.62$\pm$ 45.77|733.50$\pm$ 24.98|729.60 $\pm$ 49.87|**1147.48 $\pm$ 29.26**|
> |[Qbert](https://anonymous.4open.science/r/NECSA-ICLR-2023/rebuttal/Qbert.png)|9670.13 $\pm$ 464.31|10817.71 $\pm$ 656.88|9617.25 $\pm$ 464.31|9643.69 $\pm$ 492.48|**12659.28 $\pm$ 471.96** |
> | [MsPacman](https://anonymous.4open.science/r/NECSA-ICLR-2023/rebuttal/MsPacman.png)|1997.72 $\pm$ 120.59|2075.16 $\pm$ 106.50|1991.82 $\pm$ 78.71|1990.81 $\pm$ 134.34|**2205.28 $\pm$ 81.38** |
> | [SpaceInvaders](https://anonymous.4open.science/r/NECSA-ICLR-2023/rebuttal/SpaceInvaders.png)|629.85 $\pm$ 42.62|810.12 $\pm$ 49.93|799.06 $\pm$ 26.32|686.25 $\pm$ 53.36| **951.62 $\pm$ 28.03** |

---

> ### Author Response · Authors · 2022-11-30
> **Response to ReviewerD6zD: Experiments on Atari and DMControl (Part2)**
>
> Furthermore, we also conducted the ablation studies and hyperparameter experiments on NECSA in the following settings:
> - Ablation Study:
> 	- We compare the performance of NECSA on different multi-step analyses (1, 2, and 3 steps). The results in this [link](https://anonymous.4open.science/r/NECSA-ICLR-2023/rebuttal/Alien-steps.png) show that 3-steps is the most effective setting.
> 	- We compare the performance of NECSA using scores and average Q-values for abstract pattern measurements. The results in this [link](https://anonymous.4open.science/r/NECSA-ICLR-2023/rebuttal/Alien-scores.png) show that using scores as measurement can significantly improve the sample efficiency.
>
> - Hyperparameters:
> 	- We compare the performance of NECSA on abstracting the states and hidden outputs. To be specific, we use a fixed policy network to extract the static features of the image inputs as states, and the real-time policy network extracts the dynamic features as state-action representations. The result in this [link](https://anonymous.4open.science/r/NECSA-ICLR-2023/rebuttal/Alien-state.png) shows that using hidden outputs achieves higher sample efficiency.
> 	- We compare the performance of NECSA on different grid numbers for abstraction. The results in [link](https://anonymous.4open.science/r/NECSA-ICLR-2023/rebuttal/Alien-epsilon.png) show that five grids are the most effective choice for abstraction.
> 	- We compare the performance of NECSA with different values of epsilon. The results in this [link](https://anonymous.4open.science/r/NECSA-ICLR-2023/rebuttal/Alien-grid.png) show that 0.1 is the most suitable epsilon value for reward shaping.
> 	- NECSA is built on Rainbow for Atari tasks. We have also conducted evaluation on the performance of NECSA with different backbone algorithms. The results in [link](https://anonymous.4open.science/r/NECSA-ICLR-2023/rebuttal/Alien-backbone.png) demonstrate that (1) NECSA can achieve significant improvements in sample efficiency on different backbone algorithms and (2) the selection of backbone algorithms can affect the performance of NECSA.
>
> - The above conclusions on Atari tasks are consistent as NECSA for MuJoCo tasks. We use the same hyperparameter settings for all the Atari and DMControl tasks.
>
> Overall, we conducted large-scale and relatively comprehensive experiments on 6 Atari games to demonstrate that NECSA is general and effective for most DRL tasks. Please refer to our [program] (https://anonymous.4open.science/r/NECSA-ICLR-2023/README.md) for more details.
>
> ## DMControl Tasks with image-based states
> We tested NECSA on DMControl tasks with image-based states. We compare NECSA to DrQ-v2 [5]. DrQ-v2 is the state-of-the-art method for solving continuous control tasks with image-based states. Note that existing episodic control methods (e.g., GEM) are not applicable to continuous control tasks with image-based states, forcing us unable to compare with them. We select three continuous control tasks: Walker-Walk, Hopper-Stand, and Cartpole-Balance. All the experiments run in 500,000 training steps on five random seeds. We strictly follow the hyperparameter settings and network architecture in Appendix B of DrQ-v2 [5] to enable the fair comparison.  The implementation repository is in the anonymous [link](https://anonymous.4open.science/r/drqv2_necsa/README.md). The overall results are reported as follows:
>
> | Task Name | DrQ-v2| NECSA|
> |:----:|:----:|:----:|
> |[Walker-Walk](https://anonymous.4open.science/r/drqv2_necsa/images/Walker-Walk.png)|569.28 $\pm$ 102.80|**903.36$\pm$ 28.77**|
> |[Hopper-Stand](https://anonymous.4open.science/r/drqv2_necsa/images/Hopper-Stand.png)|380.51 $\pm$ 66.79|**856.59$\pm$ 61.62**|
> |[Cartpole-Balance](https://anonymous.4open.science/r/drqv2_necsa/images/Cartpole-Balance.png)|983.29 $\pm$ 10.66|**992.31 $\pm$ 10.58**|
>
> Our results reveal that NECSA can significantly outperform DrQ-v2 on DMControl tasks. In terms of the results on MuJoCo (in the paper), Atari, and DMControl domains, we could conclude that NECSA is effective on (1) concrete states and pixel-based states, (2) continuous and discrete action spaces and (3) a mixture of them. We believe that NECSA can be applied to most DRL tasks and improve various DRL algorithms.
>
> ## Reference
> [1] Hu, Hao, et al. "Generalizable episodic memory for deep reinforcement learning." _arXiv preprint arXiv:2103.06469_ (2021).
>
> [2] Hansen, Steven, et al. "Fast deep reinforcement learning using online adjustments from the past." _Advances in Neural Information Processing Systems_ 31 (2018).
>
> [3] Mnih, Volodymyr, et al. "Playing atari with deep reinforcement learning." _arXiv preprint arXiv:1312.5602_ (2013).
>
> [4] Hessel, Matteo, et al. "Rainbow: Combining improvements in deep reinforcement learning." _Thirty-second AAAI conference on artificial intelligence_. 2018.
>
> [5] Yarats, Denis, et al. "Mastering visual continuous control: Improved data-augmented reinforcement learning." _arXiv preprint arXiv:2107.09645_ (2021).

---

### Public Comment · ~Hao_Sun1 · 2023-02-21
**Question on Action Selection**

Dear Authors,

Congratulations and many thanks for the insightful paper! I have some questions about the action selection step: in your implementation, different baselines are combined with NECSA such that NECSA is a plug-in. I wonder if you have tried to combine it with other algorithms like SAC?

Also, I wonder if the authors can provide more details on the actions selected during online data collection. What is the exact policy used in choosing action and how is it updated?

Many thanks,
Hao

---

> ### Author Response · Authors · 2023-03-05
> **Response to Hao**
>
> 1."In your implementation, different baselines are combined with NECSA such that NECSA is a plug-in. I wonder if you have tried to combine it with other algorithms like SAC?"
>
> Thanks for the question. NECSA is a highly adaptable algorithm that can be seamlessly integrated as a plug-in into most deep reinforcement learning paradigms. Specifically, NECSA abstracts the states (-action pairs or hidden outputs) from the historical data and measures them. The measurements obtained from NECSA can re-shape the rewards without relying on a specific backbone algorithm. NECSA is compatible with various reinforcement learning algorithms such as TD3, DDPG, and SAC. However, we could not test all RL algorithms due to time constraints.
>
> 2."I wonder if the authors can provide more details on the actions selected during online data collection. What is the exact policy used in choosing action and how is it updated?"
>
> There are no action selections in NECSA. Instead, the state-action (hidden outputs) pairs are collected from the agents' and environments' interactions (the same as general DRL algorithms). Each action is generated by the policy based on the current state. Therefore, we do not perform action selection in NECSA.

---

### Decision · Program_Chairs · 2023-01-20

**Decision:**

Accept: notable-top-25%

**Justification For Why Not Higher Score:**

Contribution is focused on agents/control, so may be of interest to only a subset of ICLR attendees.

**Justification For Why Not Lower Score:**

Extensive empirical results showing a significant step forward for Neural Episodic Control justify acceptance (as opposed to rejection.)

The extensive literature review and clear introduction to methods not commonly known would be good to promote to a larger audience via spotlight presentation (as opposed to poster.)

**Metareview: Summary, Strengths And Weaknesses:**

The authors propose a method to combine state abstraction with Neural Episodic Control significantly improving sample efficiency in a wide range of typical deep reinforcement learning environments. The paper was well received throughout the review period, with many reviewers commenting positively on the extensive literature survey and clear presentation of ideas not overly common in the deep reinforcement learning literature. Some initial concerns regarding the applicability of the method to environments with visual state representations were addressed with additional experimental results in both Atari and DMControl Lab during the discussion period. Overall this paper appears to contribute both a good introduction to Neural Episodic Control and a significant step forward in the applicability of this approach to complex environments.

**Note From Pc:**

if the above contains the word "oral" or "spotlight" please see: "oral" presentation means -> notable-top-5% and "spotlight" means -> notable-top-25%. As stated in our emails, we are disassociating presentation type from AC recommendations